# Probing the atomically diffuse interfaces in Pd@Pt core-shell nanoparticles in three dimensions

Zezhou Li[1,2], Zhiheng Xie[1,2], Yao Zhang ®[1,2], Xilong Mu[1,2], Jisheng Xie[1,2], Hai-Jing Yin[1], Ya-Wen Zhang ®[1], Colin Ophus[3] & Jihan Zhou ®[1,2] ✉

Deciphering the three-dimensional atomic structure of solid-solid interfaces in core-shell nanomaterials is the key to understand their catalytical, optical and electronic properties. Here, we probe the three-dimensional atomic structures of palladium-platinum core-shell nanoparticles at the single-atom level using atomic resolution electron tomography. We quantify the rich structural variety of core-shell nanoparticles with heteroepitaxy in 3D at atomic resolution. Instead of forming an atomically-sharp boundary, the core-shell interface is found to be atomically diffuse with an average thickness of 4.2 Å, irrespective of the particle's morphology or crystallographic texture. The high concentration of Pd in the diffusive interface is highly related to the free Pd atoms dissolved from the Pd seeds, which is confirmed by atomic images of Pd and Pt single atoms and sub-nanometer clusters using cryogenic electron microscopy. These results advance our understanding of core-shell structures at the fundamental level, providing potential strategies into precise nanomaterial manipulation and chemical property regulation.

Core-shell nanoparticles (CS-NPs) are attracting significant interest in chemistry and material science, and are widely used in many applications such as catalysis, optics, electronics, and biomedical applications due to their optical, magnetic, and electronic characteristics[1–6]. The versatile functionalities of CS-NPs arise from the physical and chemical properties of both the core and the shell[7,8], determined by the three-dimensional (3D) arrangement of the atoms, particularly at the core-shell interfaces. The core and shell structures can be modified by tailoring the composition, size and morphology of each accurately, leading to a large number of combinatorial properties[9,10]. For example, by managing the surface strain[11] or designing the thickness[12] of the shell, the catalytic performance of bimetallic CS-NPs can be tuned in a controlled manner.

The characterization of CS-NPs is critical to understand structure-functionality correlations, which in turn influence the design of materials with improved properties. During heteroepitaxial growth of the shell in solution, geometric misfit can introduce interfacial strain into the CS-NPs[13], resulting in better electronic and catalytical properties on the surface[14]. It has also been reported that the formation of alloyed-shell can release the strain at the core-shell interface and enhance the luminescence quantum efficiency of colloidal quantum dots[15,16]. It is important to show how the deformation of the lattice maintains coherency across the interfaces in the core-shell heteroepitaxial structures[17,18]. Thus, obtaining fine structure of core-shell interface at atomic resolution is crucial to understand the overall behavior of CS-NPs[19,20]. In the past decades, through rapid development of electron microscopy, both structural and elemental information of CS-NPs can be routinely characterized at atomic resolution using either transmission electron microscopy (TEM) or scanning transmission electron microscopy (STEM)[21–26]. However, investigation of nanostructures from two-dimensional images could give deceptive structural information due to the overlap of critical features in the

[1]Beijing National Laboratory for Molecular Sciences, College of Chemistry and Molecular Engineering, Peking University, 100871 Beijing, China. [2]Center for Integrated Spectroscopy, College of Chemistry and Molecular Engineering, Peking University, 100871 Beijing, China. [3]National Center for Electron Microscopy, Molecular Foundry, Lawrence Berkeley National Laboratory, Berkeley, CA 94720, USA. ✉e-mail: jhzhou@pku.edu.cn

vertical dimension[27]. The 3D atomic structure of most CS-NPs, especially the atomic arrangement of the solid-solid interfaces between the core and the shell remains unknown.

To address these critical issues of core-shell interfacial structure, here we synthesized polycrystalline pentagonal bipyramid and monocrystalline truncated octahedron palladium-platinum CS-NPs using two-step chemical reduction, and then probed the 3D atomic structures of CS-NPs at the single-atom level. The 3D atomic coordinates and chemical composition of Pd@Pt CS-NPs with different morphology and size were determined using atomic resolution electron tomography (AET), which has been a useful tool for atomic-scale structural characterization in 3D[27,28,30–32] and even 4D[29]. We confirmed the heteroepitaxial growth of Pt shell on the Pd seeds based on the structural information including both the five-fold symmetry and monocrystalline CS-NPs. The 3D strain tensor analysis shows no significant difference between the interface and the whole particle. We discovered a 3D atomically diffuse core-shell interface and find the average thickness of this diffuse interface in Pd@Pt CS-NPs is 4.2 Å, irrespective of the particle's shape or crystallographic texture. The radially-averaged concentration of Pd in the interface drops in a two-sided distribution, with a smaller interface width for Pt concentration in the Pd core and a larger interface width for the Pd concentration in the Pt shell, which is correlated to the free Pd atoms dissolved from the Pd seeds. We discussed the origin of the diffuse interfaces by correlating these observations with possible growth mechanisms. These findings pave the way for further studies of nanomaterial manipulation and chemical property regulation.

## Results

### 3D atomic structures of Pd@Pt nanoparticles

Pd@Pt CS-NPs with different morphologies were synthesized by a two-step chemical reduction method[33] (Supplementary Table 1, Supplementary Fig. 1a–f, and "Methods"). We focus on three representative particles with different shapes; namely, pentagonal bipyramid (PB), elongated pentagonal bipyramid (EPB) and truncated octahedron (TO) in this study. Energy-dispersive X-ray spectroscopy (EDS) mappings

show well-defined Pd@Pt core-shell structures (Supplementary Fig. 1g and Supplementary Fig. 2a, c). Line-cuts of the EDS maps along one axis of the CS-NPs show strong Pt signal at the edge layer/layers of the shell with almost no Pd signal, indicating that the surface of CS-NPs is pure Pt without Pd. (blue arrows in Supplementary Fig. 2b, d). Tomographic tilt series (Supplementary Figs. 3–6) were acquired from several nanoparticles with different morphologies using an aberration-corrected transmission electron microscope in annular dark-field (ADF) STEM mode (Supplementary Table 2).

After pre-processing and image denoising, 3D reconstructions were computed from the tilt series using an iterative algorithm described elsewhere[30]. All the 3D atom coordinates were determined and classified ("Methods", Supplementary Table 2). To verify our reconstruction, atom tracing and classification procedures, we used multi-slice simulations to generate the same number of projections from all three experimental models including PB, EPB and TO ("Methods"). Supplementary Fig. 7a–c show the experimental and simulated projections are consistent. Then we computed new models using the same reconstruction, atomic tracing, and classification procedures. By comparing experimental models with multi-slice ones, we estimated that 99% percent of atoms were identified correctly with a 3D precision of 19 nm, 14 nm, and 17 nm for PB, EPB and TO, respectively (Supplementary Fig. 7d). PB, EPB and TO particles are different in size, with a large variety in both core and shell atom numbers. Figure 1a, b and Supplementary Movie 1 show the experimental 3D atom models of PB, EPB and TO particles with Pd atoms in blue and Pt atoms in yellow. The insets in Fig. 1a show three corresponding Johnson solids representing the shape of each particle. The sizes of these three particles are around 8.5 nm, 6.0 nm, and 7.2 nm, respectively. Despite of the variety in atom number and shape, the interfaces between the Pd core and the Pt shell show diffuse chemical profiles, without a defined sharp boundary in chemical composition for all three particles (Fig. 1c, Supplementary movie 2). The diffuse core-shell interfaces with single Pd atoms spreading into the Pt shell were observed in the internal atomic slices in all three particles (Fig. 1c). From the experimental 3D coordinates, we quantitatively

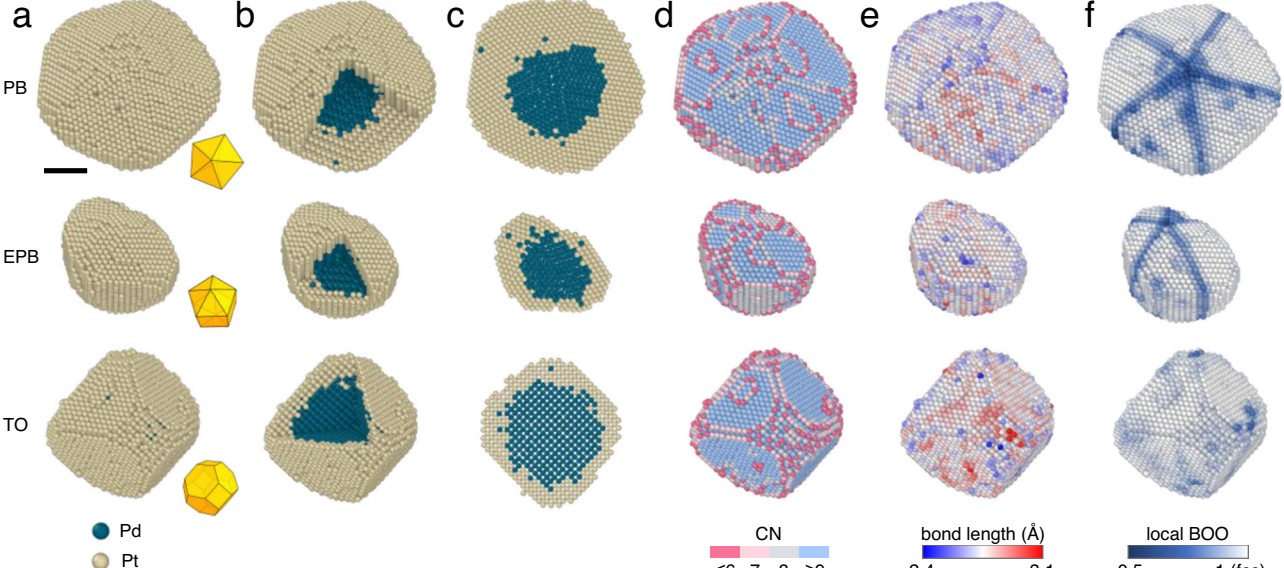

**Fig. 1 | 3D atomic structures and chemical compositions of three core-shell nanoparticles with different morphologies. a–f** 3D surface morphology (**a**), core-shell structure (**b**), internal atomic slice (**c**), CN (**d**), bond length (**e**), and local BOO parameters (**f**) of PB, EPB and TO particles were identified from the coordinates determined by AET at the single-atom level. Inserts in **a** are the geometric Johnson solid models of three particles. For the structural order in **f**, BOO = 1 corresponds to

a perfect fcc lattice. Scale bar, 2 nm. In **a–c**, green balls and yellow balls represent Pd and Pt atoms, respectively. In **d**, pink, pale pink, pale blue and blue colors represent atoms with CN ≤ 6, =7, =8 and ≥9, respectively. In **e**, continuous gradient colors from blue to red represent atoms with mean bond length from 2.4 Å to 3.1 Å. In **f**, continuous gradient colors from blue to white represent atoms with local BOO from 0.5 to 1, where BOO = 1 means ideal fcc structure.

characterized the coordination number (CN), bond length and local bond orientation order (BOO) parameters at the single-atom level (Fig. 1d–f). The CN of surface atoms at {111} facets are almost all 9, while CN values smaller than 8 are common at the edges and corners. Almost no difference was observed in the distributions of the Pd-Pd, Pt-Pt and total bond lengths (Supplementary Fig. 8a–c). The mean bond lengths of three particles were calculated as 2.76, 2.76 and 2.77 Å for PB, EPB and TO, respectively (Methods), close to the standard Pt-Pt bond length of pure Pt metal (2.774 Å)[34]. This is considerable because in all three particles Pt atoms are a few thousand numbers more than Pd atoms. We also observed a small lattice compression of the Pt shell (Supplementary Fig. 8d–f) which is attributed to the effect of surface tension and strong bonding of surface atoms[35]. We performed electron diffraction (ED) and XRD measurements on both Pd seeds and corresponding CS-NPs (Supplementary Fig. 9), and the bond lengths obtained from ED and XRD are consistent with our AET 3D measurements (Supplementary Table 3). We calculated the BOO maps using polyhedral template matching[36] (Methods). The TO particle shows a face-centered cubic (fcc) single crystal structure, whereas the pentagonal symmetric PB and EPB particles have five twined fcc grains with hexagonal close packing (hcp) like boundaries (Fig. 1f).

## Heteroepitaxial growth of Pt shell

To probe the structural information both in the interfaces and the whole particles at atomic resolution, we analyzed the point defects, grain boundaries and edges/corners in all three particles. No vacancies were found in any of the three particles. Figure 2a–c show the 3D reconstruction volumes of PB, EPB, and TO particles projected from the <110>, <110> and <100> directions, respectively. The five grains in PB are almost the same in size (Fig. 2a), while the five grains in the EPB have a large variation in size (Fig. 2b), which is due to the small immature five-fold grains in Pd core (Fig. 2e). We observed alternating {100} and {111} crystal planes with adjacent step edges connecting two neighboring facets and corners (Fig. 2a–c and Supplementary Fig. 10a–c). Instead of growing out a sharp corner at each edge, the ridges and edges in all particles form concave grooves to minimize the surface energy (Supplementary Fig. 10d–f).

Figure 2d–f show the central 2.7-Å-thick reconstruction slices of three reconstructions of PB, EPB and TO, respectively. Each facet on the surface of the particles (solid lines) corresponds to a smaller facet on the surface of Pd core (dotted lines). The morphology of the CS-NPs depends on the morphology of the Pd seeds (Pd core). Figure 2g–i and Supplementary movie 3 show the cut-outs and corresponding atomic slices of shells in three particles in <100> (yellow arrows) and <111> (blue arrows) directions. All our observations suggest the heteroepitaxial growth of the Pt shell on the Pd core, where the Pd and Pt lattice constants are close enough to maintain coherency and make the heteroepitaxial growth process energetically more favorable[20,37].

Figure 2j shows the five-fold co-axis penetrating through the core-shell interface in PB and EPB. The atoms in the axis are still twelve coordinated, but exhibit a local environment with ten hcp-ordered atoms at twin boundaries and two adjacent coaxial atoms, forming two decahedra. There exists slight distortion in each decahedron compared to an ideal Johnson polygon, and the decahedron chain is also twisted along the axis direction. The angles between twined fcc grains are measured as 74.1°, 70.8°, 71.8°, 72.3°, 71.0° for PB particle and 70.6°, 72.2°, 72.7°, 72.1°, 72.4° for EPB particle, which slightly deviate from perfect five-fold symmetry (72°)[38]. These distortions possibly alleviate the strain in the five-fold grain boundaries. Full 3D strain tensor analysis[39] shows most of all six components of the full strain tensor are around 1% (Supplementary Figs. 11–13, "Methods"), smaller than most of the strain measurement results reported[18,40–43]. No significant difference exists between the interface and the whole particle (Supplementary Fig. 14), promoting a roughly uniform growth of Pt shell on different crystallographic texture. Both our ED and XRD

results also indicate that the lattice distortion induced strain was not dominant in our CS-NPs (Supplementary Table 3).

## 3D atomically diffuse core-shell interfaces

To better visualize the internal 3D core-shell interface, we divided the atomic coordinates of three particles into slices four-atomic-layer-thick (Fig. 3a–c). The 3D interfaces between Pd and Pt are diffuse with a consistent thickness, mainly distributed in the Pt shell part. In addition, there are isolated Pd atoms and clusters scattered in the Pt shell. Figure 3d and Supplementary Fig. 15a, b show the similar distributions of isolated Pd atoms in the TO, PB and EPB particles, respectively. However, very few isolated Pt atoms are found in the Pd core (Supplementary Fig. 15d–f). One representative cut-out of the atomic model of TO particle is shown in Fig. 3e, showing mixed interface and isolated Pd atoms.

To quantitatively confirm the thickness of the diffuse core-shell interface, we calculated the concentration of chemical composition and mean CN at a depth step of 1.03 Å (Methods). The radially-averaged concentration of Pd in all three particles drops quickly from the Pd core to the center of the interface, and then decreases at a much slower rate from the center of the interface into the Pt outer shell (Fig. 3f). Since AET has been proven to achieve accurate chemical species information with 95% consistency[29], we measured the thickness of the CS interface using Pd concentration between 5% and 95% as the defined thickness range (Supplementary Fig. 15c). The thicknesses of CS interfaces in all three NPs are equal to 4.2 Å, irrespective of the morphology (Fig. 3f), roughly three-atomic-layer-thick. In addition, no significant difference is observed between the thicknesses of diffusive interfaces along <111> direction and the ones along <200> directions (Supplementary Fig. 15g–i).

To probe the Pt and Pd coordination, we analyzed the number of nearest-neighbor Pd around a Pt atom and the number of nearest neighbor Pt around a Pd atom in each layer, termed CN-$Pt_{Pd}$ and CN-$Pd_{Pt}$, respectively. CN-$Pd_{Pt}$ = 12 means a Pd site is surrounded by 12 Pt atoms, indicating that this Pd atom is isolated in pure Pt lattice. The mean CN-$Pt_{Pd}$ decreases while the mean CN-$Pd_{Pt}$ increases from core to shell (Fig. 3g), indicating that Pd and Pt atoms are diffusively mixed in the interfaces, rather than forming an atomically sharp boundary. We counted all the isolated atoms and clusters in three particles (Fig. 3h); 82.9% of them are single Pd atoms distributed in a 10-Å-thick layer in the Pt shell (Fig. 3i).

## Discussion

Although TEM/STEM has been employed to explore the growth mechanism of CS structures in 2D[10,26,44–46], it was speculated that the intermixing of core and shell atoms could happen before a perfect shell was formed[47]. Our experimental study of 3D interfaces in CS-NPs reveals several observations at the single-atom level. First, the CS interfaces are diffuse in different NPs with almost the same thickness. Second, isolated Pd atoms in the Pt shell are far more common than vice versa (Fig. 3d and Supplementary Fig. 15a, b, d–f). Third, the diffuse interface is an asymmetric where the Pt concentration drops over a short distance into the Pd core (width from 99% to 95% is 0.6 Å), while the Pd concentration drops more slowly into the Pt shell (width from 5% to 1% is 4.1 Å) (Fig. 3f and Supplementary Fig. 15c). The striking similarities among the core-shell interfaces of three particles, irrespective of the particles' morphology or crystallographic texture, suggest a potential mechanism during the wet chemical reduction to form the diffuse interfaces.

To explore the shell growth mechanism, we performed high resolution STEM imaging of the pure Pd seeds and intermediate products of TO nanoparticles using cryogenic temperature holder ("Methods"). By freezing the Pd seeds and intermediate products at liquid nitrogen temperature, we acquired the HR-STEM images of nanoparticles and their surroundings at atomic resolution (Fig. 4a–c). To better visualize the single atoms and small clusters, we saturated the

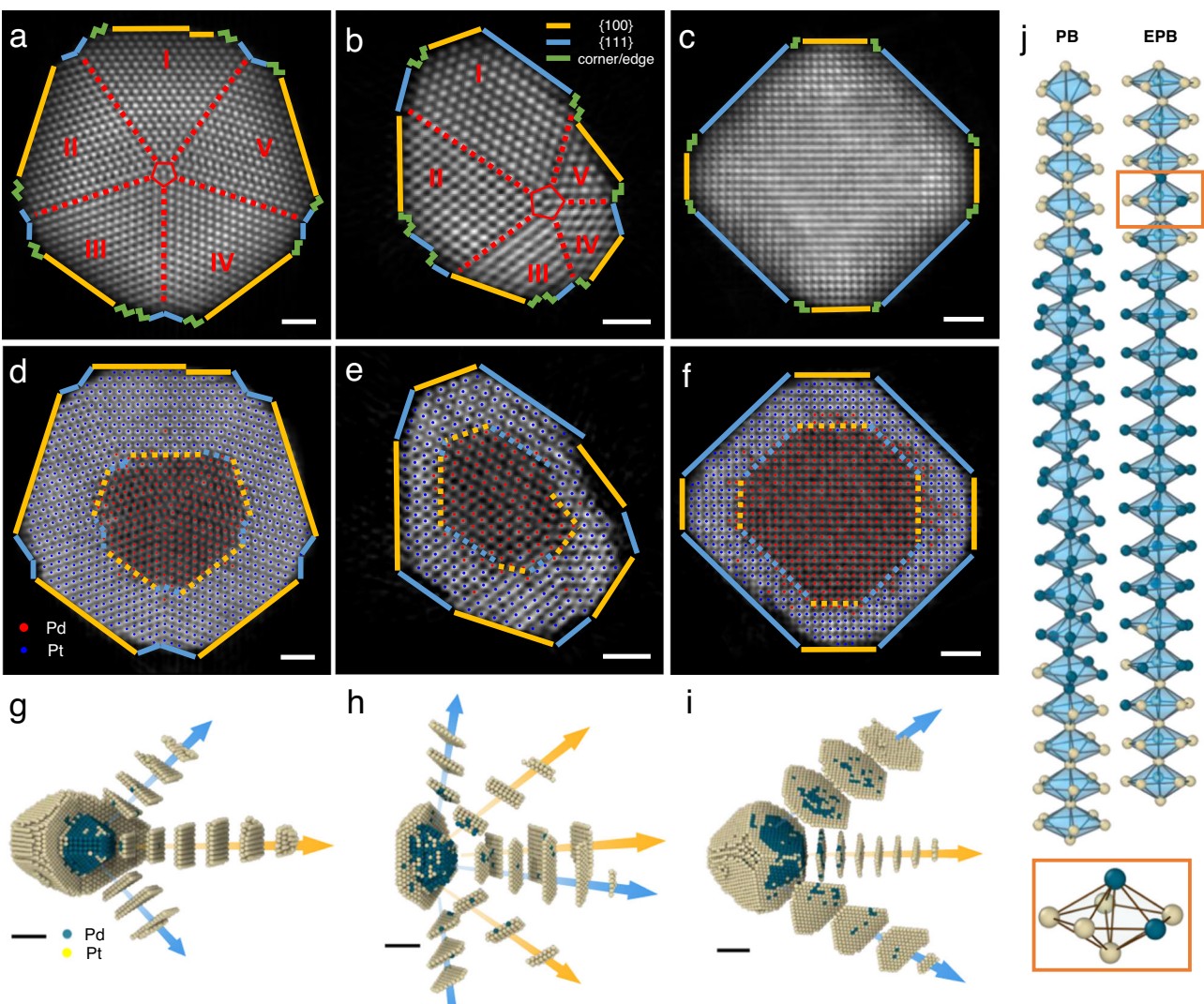

**Fig. 2 | Heteroepitaxial growth of Pt shell in core-shell nanoparticles.**
**a–c** Projections of 3D reconstruction of PB (**a**), EPB (**b**), and TO (**c**), viewed from <110>, <110> and <100> directions, respectively. For PB and EPB, <110> direction is the co-axis of five single crystal grains (five-fold co-axis) direction. **d–f** The central 2.7-Å-thick reconstruction slices of three reconstructions of PB (**d**), EPB (**e**), and TO (**f**), respectively. **g–i** Cut-out atom models of PB (**g**), EPB (**h**), and TO (**i**) with some grains of shell sliced in atomic layers, indicating heteroepitaxial growth of {100} and {111} facets from core to shell. Yellow and blue arrows show the direction of <100> and <111> directions, respectively. **j** Fivefold co-axis of PB and EPB with their nearest neighboring coordination polyhedra. All atoms in the five-fold co-axis are twelve-coordinated in a decahedron structure. Orange box enlarges a decahedron with slight distortion. Scale bar in **a–f** is 1 nm, in **g–i** is 2 nm. In **a–f**, yellow and blue lines show {100} and {111} facets parallel to the view direction. Green fold lines represent corners and steps. In **a**, **b**, the five-fold co-axis and its nearest neighbors are marked in red pentagon, and the five-fold twin boundaries are highlighted by red dashed lines, each grain labeled from I to V. In **d–f**, the red and blue dots show the coordinates of Pd and Pt atoms, respectively. In **g–j**, green balls and yellow balls represent Pd and Pt atoms, respectively. In **g–i**, yellow and blue arrows represent the direction of {100} and {111} facets, respectively.

brightness in Fig. 4d–f. The Pd seeds are surrounded by free single atoms and small atom clusters (highlighted with yellow circles in Fig. 4d), indicating that Pd seeds can dissolve in oleylamine when heated to 180 °C before adding Pt precursor. Figure 4b shows the onset of the shell growth, in which the seeds are almost the same as in Fig. 4a, except that several very bright dots (Pt atoms) are observed on the particle (red circles in Fig. 4b). Free atoms and clusters observed on the substrate are marked with yellow circles in Fig. 4e. As the Pt shell atoms grow further, a very thin layer of Pt (Fig. 4c) is observed but with Pd atoms scattered in this layer. Our experiments have shown that there is excess Pd atoms dissolved in the solvent during Pt shell growth, which is incorporated into particles until the Pd in the solvent is fully depleted.

We have examined the influence of heating procedures during sample preparation. We baked the CS-NPs at 180 °C in vacuum for much longer time than synthesis, up to 48 h. The 3D atomic models

from two independent AET measurements of the same nanoparticle before and after baking were obtained using the same procedures. By comparing their 3D atomic coordinates and species, we confirmed that 90.9% atoms in the bulk of the particles were consistent between the two models, with a chemical consistency of 97.8% (Supplementary Fig. 16a), as high as AET can possibly yield[29,48]. The same atomic slices in both models show that the diffuse Pd atoms in the Pt shell remain the same position and chemical environment (Supplementary Fig. 16b, the yellow circles). No nucleation or phase transition was observed during heating. It has been reported that the core-shell structure remains almost identical at 400 °C[49]. Thus, the diffuse Pd in the interface does not originate from the atomic motion due to the heating during the synthesis or baking process.

It is notable that we washed the Pd seeds and re-dispersed them in fresh oleylamine to get rid of any extra Pd(II) salts for the shell growth.

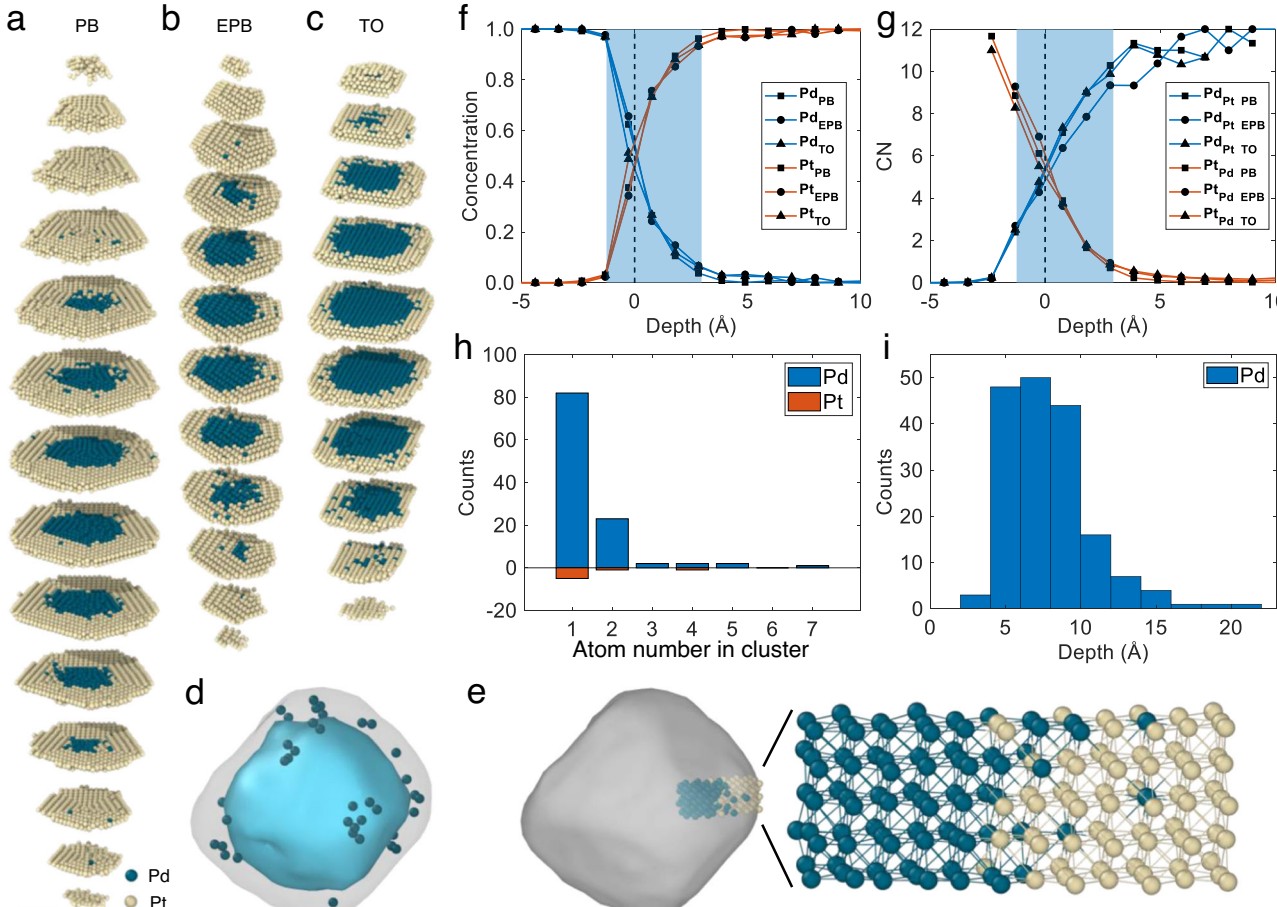

**Fig. 3 | 3D atomically diffuse core-shell interfaces in three particles. a–c** Atomic slices of PB (**a**), EPB (**b**), and TO (**c**) show diffuse core-shell interface in each particle. Scale bar for **a**–**c**, 2 nm. **d** Distribution of isolated Pd atoms in TO particle. The gray and cyan contours show the full nanoparticle and the Pd core, respectively. **e** A representative cut-out from the Pd-Pt interface in TO. **f** Radially-averaged Pd (blue lines) and Pt (red lines) concentrations along the core to the shell, where the diffuse interface is highlighted by blue patch. **g** Mean Pd CN around Pt (labeled as CN-Pt$_{Pd}$) and Pt CN around Pd (labeled as CN-Pd$_{Pt}$) in each layer. The dotted black lines in **f** and **g** mark the zero-depth position. **h** Histogram of isolated Pd and Pt atoms at the interface. **i** Distribution of the distances from the isolated Pd atoms to the middle of the interface for all particles. Zero depths in **f**–**i** are defined as where the Pd concentration for each particle meets 50%. In **a**–**e**, green balls and yellow balls represent Pd and Pt atoms, respectively.

The size distributions between Pd seeds and the final Pd cores in CS-NPs (Supplementary Fig. 17) are almost identical with a precision better than 2 Å ("Methods"), indicating any etching of Pd with Pt(IV) larger than 2 Å was negligible. It has been reported that the galvanic replacement of the seeds would yield smaller cores in the final CS-NP system[50,51], which we have not observed. The diffuse Pd atoms in the interface cannot solely come from the galvanic replacement of the Pd seeds with Pt(IV) precursors under a reducing environment, as this would result in a smaller Pd core[50–52]. Consequently, the locally dissolved Pd(II) ions or Pd atoms possibly coordinated with ligands in the diffuse layers surrounding Pd seeds must participate in the growth of the interface.

We hypothesized that the dissolved free Pd atoms could possibly serve as a reservoir of Pd(II) (most likely in Pd(II)-ligand complexes form), which was co-reduced during the growth of Pt shell. The high concentration of Pd(II) ions near the interface is highly related to the free Pd atoms dissolved from the Pd seeds. This can also explain the origin of a similar interface in the synthesis of PB and EPB particles, where instead of hexachloroplatinic(IV) acid (H$_2$PtCl$_6$) a less oxidative platinum(II) acetylacetonate (Pt(acac)$_2$) is used as the precursor of Pt shells. We show a schematic diagram for the possible formation mechanism of the diffuse interface (Fig. 5).

In conclusion, we quantified the rich structural variety of Pd@Pt core-shell nanoparticles including bond length, coordination number, local bond orientation order, grain boundary, and five-fold symmetry, all in 3D at atomic resolution. Our results confirmed the hetero-epitaxial growth of the core-shell structure. We quantitatively showed the atomically diffuse interface in CS-NPs synthesized by wet chemical reduction with 3D detail. Our results suggest that the diffuse interface is highly related to the free Pd atoms dissolved from the Pd seeds, which could possibly be tuned by manipulating the Debye length (e.g., tuning the ionic strength). This atomically diffuse feature could influence the electronic structures of the shell, and further affect the applications of core-shell structures in catalysis, optics and electronics[15,16,19,53,54]. These results will advance our understanding of structure-property relationships of general core-shell structures at the fundamental level.

## Methods
### Sample preparation
Pd@Pt core-shell nanoparticles were synthesized using two-step chemical reduction procedures published elsewhere[33,55,56]. The core-shell nanoparticle samples are prepared in the following procedure.

1. The Pd(II) solutions were heated at a certain temperature to yield Pd seeds with different morphologies. Tetrabutylammonium bromide (TBAB, 99%) and oleic acid were used for the growth of PB seeds (50 °C for 1 h). Tri-n-octyl phosphine (TOP, 90%) was used for the growth of EPB seeds (250 °C for 30 min).

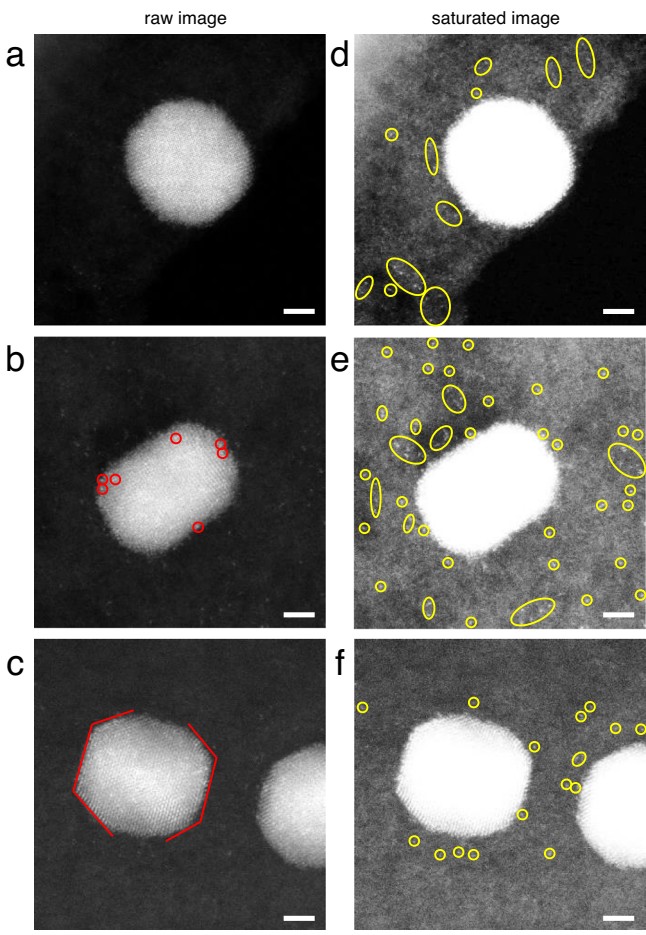

**Fig. 4 | Atomic resolution STEM images of Pd seeds and intermediate products during Pd@Pt core-shell growth in oleylamine under liquid nitrogen temperature. a** Pd seeds heated in oleylamine for 30 min with no Pt precursor added. **b**, **c** Intermediate products after Pt precursor added for 10 min (**b**) and 20 min (**c**). **d**–**f** Brightness saturated images of **a**–**c** to show the single atoms surrounding the particles. Scale bar, 2 nm. In **b**, red circles indicate the Pt atoms decorated on Pd seeds. In **c**, red lines indicate the thin Pt shell grew on Pd seeds. In **d**–**f**, yellow circles highlight free atoms and clusters surround the nanoparticles.

Poly(vinyl-pyrrolidone) (PVP, MW ≈ 55,000 g mol⁻¹), L-ascorbic acid (99.7%), KBr (99.7%) and KCl (99.7%) were used for TO seeds (80 °C for 5 min with microwave). The resulting seeds were washed by ethanol (99.7%) and re-dispersed in oleylamine (90%) by ultrasonication (Supplementary Fig. 1a–c).

2. Platinum salt solutions were added into pre-heated (180 °C) oleylamine solutions containing different seeds and aged at 180 °C for 1 h to grow the Pt shell. The final resulting CS-NPs were dispersed in cyclohexane (99.7%). HAADF-STEM images were acquired to show the particles were uniform in size and morphology (Supplementary Fig. 1d–f).

3. After deposited to 7-nm-thick silicon nitride membranes, the nanoparticles were baked at 180 °C for 24 h. in high vacuum to eliminate any hydrocarbon contamination. EDS analysis (Supplementary Fig. 1g) show that the nanoparticles were well defined core-shell structures with uniform size.

The detailed chemical formulas, purity, and concentrations of Pd and Pt precursors are listed in Supplementary Table 1.

### Particle size distribution of TO shaped Pd@Pt CS-NPs
We acquired more than fifty large-scale atomic resolution HAADF-STEM images of both TO seeds and CS-NPs. In these images, we compared the size distribution of the seeds to the size distribution of the cores in CS-NPs with a high precision. We chose TO shaped nanoparticles for statistics, since there are all in single crystalline fcc structure with rectangles shapes. Most of them were oriented along the [100] zone axis. We summarized the length and width of TO seeds and the core of CS-NPs separately from more than 200 particles by counting the number of atom columns in atomic resolution images (Supplementary Fig. 17a, b). If a particle was tilted such that only lattice fringes were visible, we measured the size only along the direction with atomic resolution. The spacing of [200] planes for fcc Pd and Pt is 1.94 and 1.96 Å, respectively. Therefore, we have estimated the size distribution of seeds and core-shells with a precision better than 2 Å (Supplementary Fig. 17c).

### Observation of core-shell growth by cryogenic electron microscopy
The CS-NPs in oleylamine solutions (180 °C) during the growth of Pt shell were dropped directly on copper TEM grids separately after heating 10 and 20 min. Then we immediately cool the grids down to near 0 °C to freeze the oleylamine. Using a cryo-TEM holder working at −193 °C (tip temperature is −179 °C), we acquired atomic resolution ADF-STEM images of nanoparticles immersed in oleylamine directly, without the severe carbon contamination problem caused by oleylamine at room temperature. Additionally, we designed a control group experiment of the TO shell growth with no Pt precursor added in. After heating the oleylamine solutions at 180 °C for 30 min, we immediately deposited and cooled the sample solution on TEM grids with the same procedure.

### Data acquisition
ADF-STEM image series with a tilt range of ±76° for three nanoparticles were acquired on an aberration-corrected FEI Titan Themis G2 300 microscope operated at 300 kV. To minimize sample drift, three sequential images were obtained with a dwell time of 2 μs in each tilt angle (Supplementary Figs. 3–5). The total electron dose of each tilt series was estimated to be between $5.6 \times 10^5$ electrons Å⁻² and $6.4 \times 10^5$ electrons Å⁻² (Supplementary Table 2). To check if the nanoparticles have structural change under electron beam irradiation, we compared the experimental zero-degree images with various forward projection images from the 3D atomic models and obtained the best matched rotation angles. Negligible structural change was observed for three nanoparticles during the tilting experiment, showing the CS-NPs were stable during the tomographic tilting imaging experiment under the total electron dose (Supplementary Fig. 6).

### Image pre-processing and 3D reconstruction
The three images were registered using normalized cross-correlation and then averaged at each tilt angle. Linear stage drift was estimated and corrected during the image registration. The experimental images were denoised using the block-matching and 3D filtering (BM3D) algorithm[57]. The BM3D parameters were optimized by minimizing the *R* factor between simulated ADF-STEM images and denoised experimental images with various denoising level. For each denoised image, a 2D mask slightly larger than the boundary of the nanoparticles was generated from each experimental image. The background level within the mask was estimated using Laplacian interpolation. After subtracting the backgrounds, images in each tilt series were aligned by the center of mass and common line method.

After image pre-processing, each experimental tilt series was reconstructed by the Real Space Iterative Reconstruction (RESIRE) algorithm[30]. Angular refinement and spatial re-alignment were applied to reduce the angular errors caused by sample holder rotation and stage instability. After no further improvement could be made, we performed the final reconstruction of each tilt series using RESIRE with the parameters shown in Supplementary Table 2.

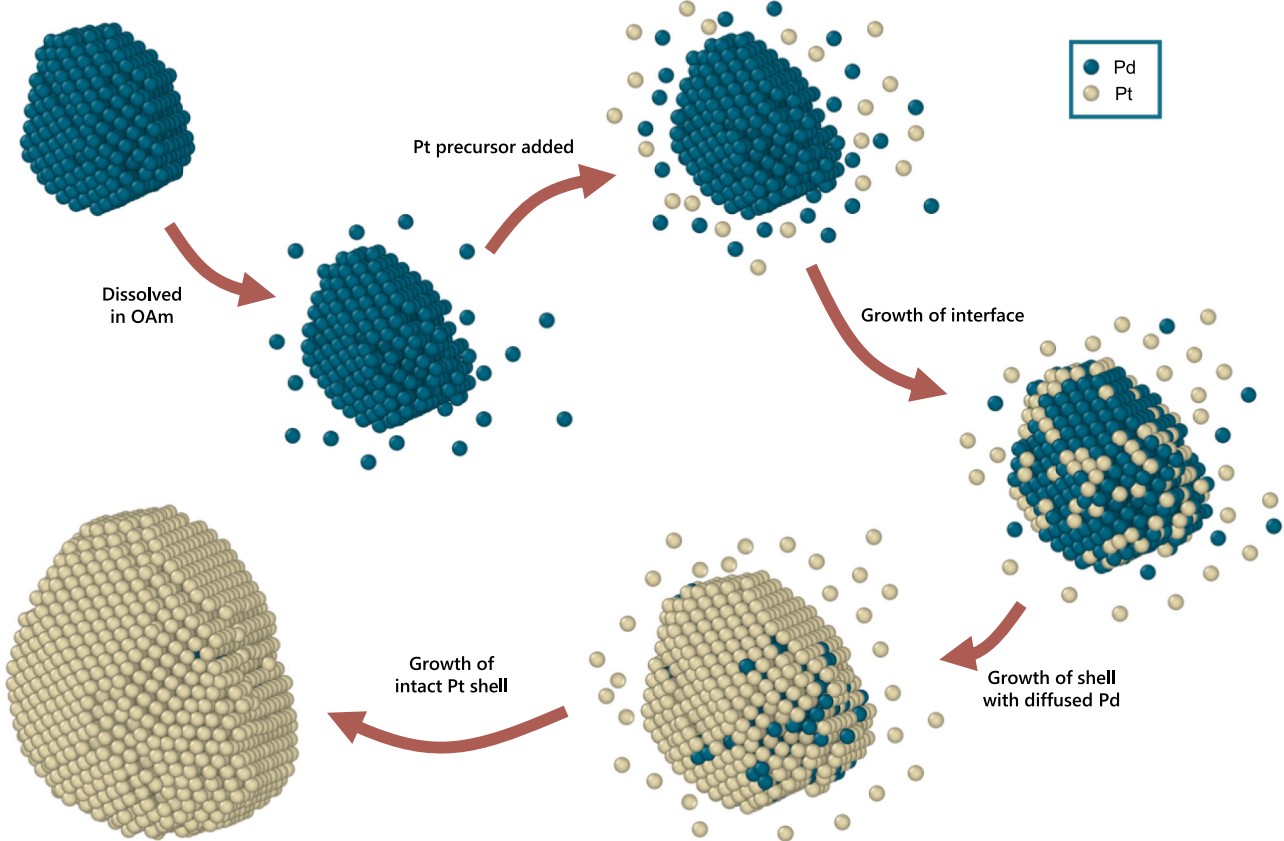

**Fig. 5 | Schematic of the proposed mechanism of core-shell growth for Pd@Pt CS-NPs.** The Pd seed was dispersed in oleylamine. A few surface Pd atoms was ionized and formed an electric double layer around the seed. After the Pt precursor was added to the solution, Pt and Pd ions were reduced simultaneously on the surface of the Pd seed to form the diffuse Pd-Pt interface with some possible substitution of Pd(0) to Pd(II) by Pt(IV). After the core was completely wrapped by Pt atoms, the shell continued to grow heteroepitaxially on the core to form the final particle. Here the atomic model of particle EPB is used for illustration. Green balls and yellow balls represent Pd and Pt atoms, respectively.

## 3D atomic coordinates determination and chemical species classification

3D atomic coordinates of three nanoparticles were determined separately using the following procedure:

1. All local maxima in the 3D reconstruction were identified and the peak positions were located using polynomial fitting[58] performed on a 7×7×7-voxel volume around each local maxima. Then we excluded non-atoms based on two criteria, i.e., integrated intensity of a box centered on the local maximum position and minimum distance between two reasonable atoms. A minimum distance constraint of 2.2 Å was used to obtained a list of potential atoms. After deleting all non-atom positions, we executed 3D polynomial fitting on atom positions to get sub-pixel resolution. About 99% of atoms were successfully traced within this procedure.

2. The 3D reconstruction always suffers from the "missing wedge" problem, the artifacts and noises in the final 3D tomography reconstruction would generate redundant local maxima on one atom area. Therefore, auto-tracing procedure will cause fitting failure and cannot yield a reasonable atom position from a large chunk of connected intensity blobs. These areas are determined to "unidentified or misidentified atoms". We further checked these positions manually to obtain proper atom positions in these areas[59].

3. Using a $K$-mean clustering method[60] based on the integrated intensity of a $2.4 \times 2.4 \times 2.4$ Å$^3$ (7×7×7 voxels) volume centered at each potential atom position, all potential atoms were classified into Pd and Pt. Due to the missing wedge problem, the intensity of the surface atoms from missing wedge direction are weaker than the atoms insides. These atoms are misclassified to Pd when using the K-mean algorithm[30]. We found they are inconsistent atoms. Our EDX mapping results clearly show Pt signal are dominate and there are no Pd atoms on the particle surface after the growth of a thick Pt shell (Supplementary Fig. 2). We manually checked a small percentage of the inconsistent atoms on the surface and performed local surface re-classification[29] at a lower intensity threshold to identify the surface misclassified Pt.

The structure of two five-fold twinned nanoparticles is analyzed by two methods, i.e., local bond orientational order (BOO) parameters and polyhedral template matching[36]. Averaged local BOO parameters ($Q_4$ and $Q_6$) and normalized BOO parameter for ideal face-centered cubic (fcc) structure are calculated based on the procedure published elsewhere[31], using the first-nearest-neighbor shell distance (3.35 Å) as a constraint. By applying polyhedral template matching, PB/EPB particle was separated into five fcc grains and a five-fold coaxial twin boundary which locally possessing hcp-like environment.

## The 3D precision estimation

We performed ADF-STEM simulations using the multi-slice method and estimated the precision of atomic coordinates by AET reconstruction[61,62]. The multi-slice simulations of tomographic tilt series for PB, EPB and TO models were performed under the same conditions as the AET experiments: 300 keV acceleration energy, −398 nm C3 aberration, 414 μm C5 aberration, 30 mrad convergence semi-angle, and 39.4 mrad and 200 mrad HAADF detector inner and outer semi-angles. 4 frozen phonon configurations and 2 Å slice thickness was

applied for the simulations. For each model, three representative multi-slice ADF-STEM images is shown, which is in good agreement with the corresponding experimental image (Supplementary Fig. 7a–c).

Using the same reconstruction RESIRE algorithm[30], 3D reconstructions were obtained from simulated tilt series images. The atomic coordinates of three particles were traced from corresponding 3D reconstruction volumes using the same procedure described in former section. We identified common atom pairs between the atomic coordinates traced from multi-slice simulated tomograms and the experiment models using half of the identical fcc Pd-Pd bond length (1.38 Å) as a threshold distance. RMSD of all common atom pairs were 19 pm (PB) with 99.8% atoms traced, 14 pm (EPB) with 99.9% atoms traced and 17 pm (TO) with 99.8% atoms traced, respectively (Supplementary Fig. 7d).

### Calculation of mean bond length
Firstly, we calculated the pair distribution function (PDF) of an atomic model or selected atomic coordinates. Then we fitted the first peak of the PDF to a Gaussian distribution function, which served as the mean bond length. The results were shown in Supplementary Fig. 8a–c.

### Displacement field and strain map calculation
We calculated the 3D displacement fields and strain maps by assigning the experimental 3D atomic model to an ideal fcc Pd lattice (the bond length is 2.750 Å)[34] in the following procedure:

1. For single crystalline particle like TO, we used iterative closest point (ICP) algorithm[63] to minimize the root-mean-square-deviation (RMSD) of the atomic model and ideal lattice. For fivefold symmetric particles like PB and EPB, we first segmented the particle into five single crystalline regions and then fitted each region separately. Upon all atom coordinates were fitted, we calculated the displacement for each atom in the particle. Atoms that fell within a quarter of the ideal bond length (0.69 Å) relative to corresponding fcc lattice sites were marked as successfully assigned atoms. For PB, EPB, and TO particles, nearly 100% atoms were fitted to ideal lattice sites.
2. Next, 3D displacement field were calculated by the interpolating the atomic displacements onto a cubic grid using kernel density estimation[39]. Here we chose a 3D Gaussian kernel with $\sigma = 3.89$ Å (ideal lattice constant of pure fcc Pd)[34] to produce a smooth estimate of the displacement field (Supplementary Figs. 11b, 12b, and 13b).
3. Finally, we calculated the 3D strain tensor by numerical differentiation of the 3D displacement field (Supplementary Figs. 11c, 12c, 13c, and 14).

### Analysis of diffusive core-shell interfaces
The diffusive core-shell interfaces were analyzed in the following steps.
1. Using the constraint that CN of Pd-Pd was not less than 12, a clean Pd core was exfoliated without any adjacent Pt atoms. Using alpha shape algorithm, a contour was calculated and smoothed based on this Pd core. Starting from this contour as an initial 3D mask, we exfoliated or dilated the mask by three-voxel-thick (1.03 Å) each time. In this way the total particle was divided layer by layer like an onion. The maximum depth was 34.0, 24.7 and 24.7 Å (33, 24, 24 layers) for PB, EPB and TO particles, respectively.
2. For atoms in each layer, the relative concentration of Pd and Pt element was determined, and the mean CNs of Pd-Pt and Pt-Pd were calculated. In this way the Pd concentration-depth line for each particle was obtained. We defined the zero-depth position as where the interpolated Pd concentration curve meets 50%. Then three curves were aligned together and averaged. We interpolated the averaged curve of Pd concentration and defined the averaged thickness of the diffusive interface with Pd concentration between 5% to 95% (Supplementary Fig. 15c).

### Reporting summary
Further information on research design is available in the Nature Portfolio Reporting Summary linked to this article.

## Data availability
All experimental data, 3D tomography reconstructions, and determined atomic coordinates are available on GitHub and Zenodo[64]. The data that support the findings of this study are also available from the corresponding author upon request.

## Code availability
Source codes are available on GitHub and Zenodo[64].

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

## Acknowledgements

This work was supported by the National Natural Science Foundation of China (Grant No. 22172003 for J.Z., 21832001 & 22293042 for Y.-W.Z.) and High-performance Computing Platform of Peking University. We thank the Electron Microscopy Laboratory and the Analytical Instrumentation Center at Peking University for the use of the aberration-corrected electron microscopes. H.-J.Y. and Y.-W.Z. thank the support of

National Key R&D Program of China (No. 2021YFA1501100) and Beijing National Laboratory for Molecular Sciences (BNLMS-CXXM-202104). Work at the Molecular Foundry was supported by the Office of Science, Office of Basic Energy Sciences, of the U.S. Department of Energy under Contract No. DE-AC02-05CH11231.

## Author contributions

J.Z. conceived the idea and directed the study. Z.L. performed the AET experiments. X.M., Z.L., J.X., H.-J.Y., and Y.-W.Z. and J.Z. discussed/ synthesized the Pd@Pt core-shell nanoparticles. Z.L., Z.X., Y.Z. and J.Z. discussed/performed image reconstruction and atom tracing. Z.L. and J.Z. analyzed the data and interpreted the results. Z.L., Y.Z., C.O., and J.Z. discussed/analyzed the 3D strain analysis. Z.L., X.M., and J.Z. wrote the manuscript. All authors commented on the manuscript.

## Competing interests

The authors declare no competing interests.
