## [Peer Review File · Nature Communications]

Probing the atomically diffuse interfaces in Pd@Pt core-shell nanoparticles in three dimensionsReviewers' Comments:

Reviewer #1:

Remarks to the Author:

Recommendation: Minor revision

The authors present a comprehensive 3D atomic structural analysis of Pd/Pt core/shell nanoparticles and the Pd-Pt interfaces through the atomic electron tomography, an emerging technique for solving 3D atomic structures of nanomaterials. The interfacial structure of a core/shell nanoparticle can be of great interest, because it governs the physical and chemical properties of the nanoparticle by, for example, altering strain distributions and defect structures. On the basis of their observations, the authors further suggested a mechanism for the core-shell nanoparticle growth. Overall, I support a publication of this manuscript. However, it is recommended to address a few concerns that I expect to make the work more attractive when it is published in Nature Communications. The suggested mechanism is too primitive and the authors should provide more rational evidences to consolidate this mechanism. In addition, the method to acquire 3D atomic coordinates should be more validated and precisely described. We have a set of questions and comments on the data and methodologies in this study as follow:

- 1) The authors argued that the oleylamine-passivated surface atoms of the Pd nanoparticle seeds are dissolved into the solvent (oleylamine) and re-deposited onto the surface with Pt during the Pt shell growth stage. However, the metal-metal binding energy is expected to be stronger than the solvation energy of Pd ion by the solvent plus ligand-solvent solvation energy. Can the authors corroborate the presence of free Pd species dissolved in the solvent?
- 2) Following the previous comment, I am skeptical on the discussion on the Debye screening length in this nanoparticle growth system. The Pt shell growth stage utilizes oleylamine, an organic molecule, as a solvent. The Pt precursors would not ionize in this solvent and therefore less likely to form electrical double layer. I believe that the authors should provide alternative mechanism to explain their observation. The authors can consider investigating intermediate products (aliquots) during the shell growth.
- 3) The suggested interface thickness, ~ 4.2 Å, is in fact relatively very thin considering the lattice parameters of Pd and Pt. If the surface of the initial seed nanoparticles is rough, this value is likely to occur without the proposed mechanism. I recommend the authors to check the surface structures of the three types of Pd nanoparticles from TEM images. Only the low-magnification images of cubic Pd nanoparticles are currently included in the Supplementary Information.
- 4) In Supplementary Fig. 8–10, the authors presented strain distributions and argued that there is no significant difference between the interface and entire nanoparticle. Previous investigations including the recent publication by Y. Yang et al. have shown the evident strain difference between core and shell regions (Small 13, 1603423 (2016); Nat. Commun. 13, 5957 (2022)). Can the authors further describe why their strain maps do not show such the different between the core and shell? In addition, I recommend the authors to include XRD or ED, and atomic-resolution HAADF-STEM or HRTEM images with measured lattice distances to show that their nanoparticles do not exhibit the strain difference between the core and shell. Finally, the authors should describe in the method section how they measured atomic displacements, strains, and mean bond lengths.
- 5) $\{111\}$ and $\{200\}$ surfaces have different surface energies. This suggests that the thickness of intermixed layer can be different depending on the facets. Any comment on this?
- 6) The authors presented three atomic maps of core-shell nanoparticles, but I think the 3D maps are not validated enough. Here, I suggest some methods to validate the maps: 1) Simulate STEM images from reconstructed maps, and compare them with the original ones; 2) Calculate the 3D precision (root-mean-square-deviation) by re-reconstruction. (Please refer to [Y. Yuan et al., Nature Materials 21, 95-102 (2022)] or [B. H. Kim et al., Science 368, 60-67 (2020)])
- 7) In 'Methods' section, the procedure for 3D atomic coordinates determination and chemical species classification is described. In the manuscript, there are some manual check and modifications as follows: "These positions were further checked manually to correct for unidentified or misidentified

atoms due to fitting failure or a large chunk of connected intensity blobs from multiple atoms"; "we performed local surface re-classification and manually checked a small percentage of the inconsistent atoms²⁷." Could you describe more on those 'manual check'? What are the criteria to determine 'unidentified or misidentified atoms' and 'the inconsistent atoms'?

8) In Fig. 2, the zone axes of (a) and (d) are different. I recommend to change Fig. 2d with an image with the same projection direction of Fig. 2a. Presenting the atomic distribution around the five-fold co-axis could be helpful for material scientist who are interested in polyhedral core-shell nanoparticles.

Reviewer #2:

Remarks to the Author:

In this manuscript, the authors revealed the three-dimensional atomic structure of Pd@Pt core shell nanoparticle by using atomic resolution electron tomography. The chemical compositions, bond length, and strain have been quantified. The diffused core shell interface has been analyzed. The following comments should be considered for revision:

For the core-shell nanoparticle growth, the shell growth is strongly reliant on core particle size and shape. Authors should provide the HRTEM/STEM of the as-synthesized core nanoparticle accompanied by the core-shell nanoparticle in Figure S1. Authors should indicate the size of core-shell nanoparticle in the 1st paragraph of the Results section.

On page 5, the authors indicate that "Despite of the variety in size and shape, the interfaces between the Pd core and the Pt shell show diffuse chemical profiles, without a defined sharp boundary in chemical composition for all three particles." In fact, the core-shell nanoparticle synthesized in this manuscript is around 8.7 nm, 6.6 nm, 7.5 nm, the word "variety" can not be used here. In a representative example [J. Am. Chem. Soc. 2015, 137, 47, 15036-15042], Pd@Pt nanoparticle is synthesized with 20 nm size, but nanoparticle > 10 nm is not included in this study. Therefore, authors may not draw the conclusion in the Introduction part "core-shell interface is atomically diffuse with an average thickness of 4.2 Å, irrespective of the particle's size, morphology or crystallographic texture"

In Figure 2 (d), EPB (e) and TO (f), the authors use the red and blue dots to show the Pd and Pt atoms, respectively. We notice that the Pd core is enclosed in a Pt shell, with some Pd atoms diffuse across core-shell interface, but there is no Pd atom at the surface. Because the catalyst properties are highly correlated to the surface atom, and authors also predict that "atomically diffuse feature affect applications of core-shell structures in catalysis." We suggest authors perform EDX elemental mapping for a single Pd@Pt nanoparticle to confirm that the surface nanoparticle is pure Pt without Pd.

On page 4, the authors indicate that "The radially-averaged concentration of Pd in the interface drops in a two-step manner, which is correlated to the electric double layers." The author should provide definitions here to explain the terminology "electric double layers". In addition, on page 14 and Figure 4, the authors indicate that "A few surface Pd atoms was [should be "were"] ionized and formed an electric double layer around the seed." This is not supported by solid data in this manuscript. In the conclusion section, the authors indicate that "Our results suggest that the diffuse interface is highly related to the electric double layers of the seeds, which could possibly be tuned by manipulating the Debye length." However, the authors did not show related data and even the method for tuning the Debye length.

Reviewer #3:

Remarks to the Author:

Li et.al, reported detail characterization of palladium-platinum core-shell nanoparticles using atomic resolution electron tomography. The authors observed an atomically diffused core-shell interface with an average thickness of ~ 4.2 Å instead of a sharp edge in all three types of Pd@Pt core-shell particles, and suggest that the diffused interface is highly related to the electric double layers of the seeds based on the correlation of the radially-averaged Pd concentrations away from the Pd core and the expected potential development of the double layers. The authors did a good job on data collection and analysis. However, the discussions to support the proposed mechanism of the core-shell growth for Pd@Pt CSNPs is not conclusive due to the following concerns, which need to be addressed before consideration of publication.

The authors conclude from Fig. 3d and Supplementary Figs. 12a-b that, isolated Pd atoms in the Pt shell are far more common than isolated Pt atoms in the Pd core. If true, this would be somewhat consistent with the proposed growth mechanism. However, please note that Fig. 3d and Supplementary Figs. 12a-b are distributions of isolated Pd atoms only outside of a 3D pure Pd mask. The comparison to isolated Pt distribution is not clear since these data are not given. Also, within the confinement of this discussion the shell and core need to be better defined. The 3D mask is generated under a subjective constrain (according to the Method) that CN of Pd-Pd was not less than 12, therefore, it's a clean Pd core without any adjacent Pt atoms by subjective choice of atomic segregation. A similar 3D mask could be made for Pt and end up with a clean Pt shell as well. Therefore there should be 3 layers using this approach: pure core, diffusion layer and pure Pt shell. If they begin the data analysis with Pt, an opposite conclusion could be drawn. The suggestion that Pt is far less likely to diffuse into Pd core is not conclusive based on these arguments yet, at least not from a logical stand point of view.

Page 12, second paragraph: "The same atomic slice in both models shows that the diffused Pd atoms in the Pt shell remain the same position and chemical environment (Supplementary Fig. 13, the yellow circles)."

The authors concluded that the diffused Pd at the interface does not originate from the atomic motion due to heating during the synthesis or baking process. However, it's unclear how thoroughly the influence of heating is examined. The discussion and figure provided seem to indicate that only one slice of each of the reconstructed 3D model from one particle before and after 48 hours of 180 °C backing is examined. If this is the case, it would be a very small sampling size to claim that the diffuse behavior remains the same for one entire particle, let alone for all particles. Also, it is not clear in the manuscript if the particle had undergone baking at 180 °C for 24 hrs in high vacuum (described in methods) before the 48 hour baking. Have the authors considered the possibility that the diffusion being initiated by baking, but stabilized after a given amount of time under constant temperature, or only when backing in solution?

Page 13, "The size distributions between Pd seeds and the final Pd cores in CS-NPs (Supplementary Figs. 1d-f) are almost identical, indicating that the etching of Pd seeds with Pt(IV) was negligible. The diffuse Pd atoms in the interface cannot solely come from the galvanic replacement of the Pd seeds with Pt(IV) precursors under a reducing environment, as this would result in a smaller Pd core" Based on the data the thickness of the diffusion layer is ~ 4.2 Å. Base on the concentration distribution curve, the etching depth, if existed, seems to be slightly less than ~ 0.2 nm. The minimum scale of the particle size statistics in supplementary Figs. 1d-f is about 0.5 nm. The resolution of the particle size statistics is maybe enough to claim that any etching of Pd larger than 1nm is negligible, but not enough to show that a 0.2 nm deep surface etching did not happen. Therefore, this data is not enough to support the claim that the diffused Pd is not solely coming from the galvanic replacement of the Pd seeds.

Unless these concerns are addressed, the only result supporting the proposed mechanism seems to be the subjective two step declining pattern of the Pd concentration curve. It will then need to be

compared with all other possible growth mechanisms to eliminate their possibility of a similar or better match.

Other minor comments:

Abstract: Instead of forming an atomically-sharp boundary, the core-shell interface is atomically diffuse with an average thickness of 4.2 Å, irrespective of the particle's size, morphology or crystallographic texture.

This claim is misleading. Analysis of diffusive core-shell interfaces was done on three different particles. Size difference of these three particles does not cover the entire possible size distribution spectrum. Claiming from this result with three data points that the diffuse layer is constant irrespective of size seems premature.

Method: The Pd(II) solutions were heated at a certain temperature to yield Pd seeds with different morphologies.

Please specify the temperatures used.

Supplementary Fig. 1f

How is the diameter defined for non-spherical particles?

Page 4: The radially-averaged concentration of Pd in the interface drops in a two-step manner, which is correlated to the electric double layers.

While the diffusion behavior and the potential near the electric double layer could be somewhat correlated, it's a stretch to conclude from observation of the radially-averaged concentration curve that the concentration drops in "two" steps.

Methods: Analysis of diffusive core-shell interfaces.

Could you explain why a fixed distance is used to exfoliate the 3D mask for the core instead of adjacent atomic layers? For different surfaces, the distance between adjacent atomic layers would be different. By fixing the distance to exfoliate, it looks like the exfoliated layer may be centered for the atoms of 1 surface layer but will cut into the edge of atoms on a different surface or may never intercept with the atomic center. How is the number of atoms determined for each layer if this is the case?

Method: Platinum salt solutions were added into pre-heated (180 °C) oleylamine solutions containing different seeds to grow the Pt shell. The solutions were aged at 180 °C for 1 hrs.

1 hrs should be 1 hr.

Response to the referees:

We thank three referees for reviewing our manuscript and making very good suggestions. After several weeks of intensive work, we have made a major revision to our manuscript and carefully addressed all the points raised by the referees and the editorial concern. By fully incorporating referees' suggestions, we believe that our revised manuscript is significantly improved. We also will provide all the data and codes on GitHub

(<https://github.com/live-long-and-prosper/PdPtCoreShellNanoparticles>) for the community. Below is a point-by-point response to the referees' comments and suggestions. All the revisions made in the manuscript are marked blue.

Response to referee 1:

Comment 1: *“Overall, I support a publication of this manuscript. However, it is recommended to address a few concerns that I expect to make the work more attractive when it is published in Nature Communications. The suggested mechanism is too primitive and the authors should provide more rational evidences to consolidate this mechanism. In addition, the method to acquire 3D atomic coordinates should be more validated and precisely described.”*

We thank referee 1 for sharing our view that this is an important work and deserves to be published in Nature Communications. We have fully addressed all the constructive suggestions made by referee 1. We performed more experiments and analysis (Fig. R1-R5, R11&R16) to support our suggested mechanism proposed in Fig. 5 in revised manuscript. We did multi-slice simulations for all three nanoparticles and validated our reconstruction and methods (Fig. R8). We added more details to the Methods section in revised manuscript, including how we measured particle size distribution, mean bond length, atomic displacements, strains, 3D precision and how we performed cryogenic electron microscopy experiments and manual tracing.

Comment 2: *“The authors argued that the oleylamine-passivated surface atoms of the Pd nanoparticle seeds are dissolved into the solvent (oleylamine) and re-deposited onto the surface with Pt during the Pt shell growth stage. However, the metal-metal binding energy is expected to be stronger than the solvation energy of Pd ion by the solvent plus ligand-solvent solvation energy. Can the authors corroborate the presence of free Pd species dissolved in the solvent?”*

This is a very important point. To further clarify the presence of free Pd species, we performed high resolution STEM imaging of the Pd seeds using cryogenic temperature holder. After we washed the Pd seeds and re-dispersed them in fresh oleylamine to get rid of any extra Pd(II) salts, we heated them to 180 °C for half an hour. Then we deposited the Pd seeds in oleylamine solution onto a copper TEM

grid. To avoid electron beam damage to the single atoms and overcome the severe carbon contamination brought by oleylamine, we facilitate HR-STEM imaging with a cryo-TEM holder (the tip temperature is at $-179\text{ }^{\circ}\text{C}$). By freezing the Pd seeds under near liquid nitrogen temperature (Fig. R1a), we acquired the HR-STEM images of Pd and its surroundings at atomic resolution. Fig. R1b show the Pd seeds are surrounded by free single atoms or atom clusters (yellow circles). This is strong evidence Pd seeds can dissolve in oleylamine when heated to $180\text{ }^{\circ}\text{C}$.

Fig. R1. Observation of Pd seeds in oleylamine under liquid nitrogen temperature. a, Large scale image of Pd seeds in frozen oleylamine. b, Atomic resolution STEM images of Pd seeds frozen in solidified oleylamine. Right rows of images in b are saturated in brightness in order to show free atoms clearly. Some free

atoms and clusters are circled yellow to facilitate observation.

Comment 3: *“Following the previous comment, I am skeptical on the discussion on the Debye screening length in this nanoparticle growth system. The Pt shell growth stage utilizes oleylamine, an organic molecule, as a solvent. The Pt precursors would not ionize in this solvent and therefore less likely to form electrical double layer. I believe that the authors should provide alternative mechanism to explain their observation. The authors can consider investigating intermediate products (aliquots) during the shell growth.”*

We followed the referee 1’s suggestion and investigated the intermediate products during the shell growth. After we added the Pt precursors into washed Pd seeds and heated at 180 °C for 10 and 20 mins, we prepared two TEM samples by pipetting the aliquots separately. We acquired the HR-STEM images of the intermediate products in frozen oleylamine under liquid nitrogen temperature. Fig. R2a shows the onset of the shell growth: in which the seeds are almost the same as in Fig. R1b, except that several very bright dots (Pt atoms) are observed on the particle (red circles in Fig. R2a). Free atoms and clusters can be easily observed near the substrate (yellow circles in Fig. R2a). As the Pt shell atoms grow further, a very thin layer of Pt (red lines in Fig. R2b) is observed but with residual free atoms scattered in this layer.

In summary, we observed the seeds with single Pd atoms, the very onset of shell growth and the intermediate core-shell structures with a very thin shell composed of both Pd and Pt (Fig. R1&R2). This corresponds to the schematic diagram we proposed in our manuscript. These images of pure Pd seeds and intermediate products validate our proposed mechanism. A new figure including these results have been updated in the revised manuscript as Fig. 4. New sentences “To explore the shell growth mechanism, ...until the Pd in the solvent is fully depleted.” (page 13-14) have been added.

raw image

saturated image

a

b

Fig. R2. Atomic resolution STEM images of intermediate products during the shell growth. **a**, Pd particles growing the Pt shell in oleylamine for 10 mins. Left panels are images in original contrast and brightness, showing Pt atoms decorated on Pd seeds marked by red circle. Right panels are images with saturated brightness, highlighting free atoms and clusters marked by yellow circle. **b**, Pd particles growing the Pt shell in oleylamine for 20 mins. Left panels are images in original contrast and brightness, showing thin Pt shells grew on Pd seeds marked by red lines. Right panels are images with saturated brightness, highlighting remained free atoms and clusters marked by yellow circle. Scale bar for each image is 2 nm.

Comment 4: “The suggested interface thickness, ~ 4.2 Å, is in fact relatively very thin considering the lattice parameters of Pd and Pt. If the surface of the initial seed nanoparticles is rough, this value is likely to occur without the proposed mechanism. I recommend the authors to check the surface structures of the three types of Pd nanoparticles from TEM images. Only the low-magnification images of cubic Pd nanoparticles are currently included in the Supplementary Information.”

Fig. R3 | Atomic resolution HAADF-STEM images of PB, EPB, TO core-shell particles and corresponding Pd seeds. **a-c**, Pd seeds of PB, EPB and TO CS-NPs, respectively. **d-f**, PB, EPB and TO core-shell nanoparticles, respectively. Scale bar, 5 nm.

This is a good point. We have followed referee 1’s suggestion and acquired several atomic resolution images of initial Pd seeds with different morphology. Overall, the surfaces of the seeds are uniform without steps or islands on crystal faces (Figs. R3 a-c). We made a new supplementary Fig. 1 showing the atomic resolution images of

PB, EPB, TO core-shell particles and corresponding Pd seeds. We want to clarify that the 4.2 Å is a definition we used to describe the averaged thickness of the diffusive interface with Pd concentration between 5% to 95%. There are very few Pd atoms (Pd concentration below 5%) scatter between 4.2 Å to ~8.3 Å in the Pt layers. We have not included these atoms when defining the averaged thickness. We clarify it in our revised Methods and Supplementary Fig. 15c.

Comment 5: “In Supplementary Fig. 8–10, the authors presented strain distributions and argued that there is no significant difference between the interface and entire nanoparticle. Previous investigations including the recent publication by Y. Yang et al. have shown the evident strain difference between core and shell regions (Small 13, 1603423 (2016); Nat. Commun. 13, 5957 (2022)). Can the authors further describe why their strain maps do not show such the different between the core and shell? In addition, I recommend the authors to include XRD or ED, and atomic-resolution HAADF-STEM or HRTEM images with measured lattice distances to show that their nanoparticles do not exhibit the strain difference between the core and shell. Finally, the authors should describe in the method section how they measured atomic displacements, strains, and mean bond lengths.”

This is a good point. We have followed referee 1’s suggestion and performed more experiments including ED, XRD and atomic-resolution HAADF-STEM to verify our strain measurement; and we compared our results with the results in literature (Y. Yang et al. *Nat. Commun.* **13**, 5957 (2022)). Our measurements showed the strain are within 5% in our CS-NPs in the manuscript, and we validated our measurement with three major methods as below.

Table R1 | Measured bond length by powder X-ray diffraction and electron diffraction.

Bond length (Å)	composition	Particle morphology	Relative Deviation Pt/Pd-1 (%)	Method
2.779	Pd	PB	0.29	ED
2.787	Pd@Pt			
2.767	Pd	EPB	0.58	
2.783	Pd@Pt			
2.746	Pd	TO	0.8	
2.768	Pd@Pt			
2.739	Pd	TO	1.3	XRD
2.774	Pd@Pt			
2.750	Standard fcc Pd		0.87	
2.774	Standard fcc Pt			

First, ED and XRD. Fig. R4a shows the ED patterns taken from all three seeds and corresponding CS-NP, typical diffraction rings of multi-grain NPs are observed. We quantitatively analyzed the radial intensity distribution as a function of distance and

calculated the averaged bond length of seeds and CS-NP for three kinds of morphology, which are summarized in Table R1. Comparing our results with the standard Pd and Pt, the differences are less than 2%, within the error of ED measurement. Additionally, our XRD measurements show that the bond lengths of TO seeds and CS-NP are very close (Fig. R4c), consistent with our AET 3D measurements. Both our ED and XRD results indicate that the lattice distortion induced strain is not dominant in our CS-NPs.

Fig. R4 | ED and XRD spectrum of PB, EPB, TO core-shell particles and corresponding Pd seeds. **a**, Electron diffractions (EDs) of Pd seeds and Pd@Pt core-shell nanoparticles for PB, EPB and TO particles, respectively. **b**, Radial averaged intensity profiles of EDs in **a** for three kinds of particles. **c**, XRDs of TO seeds and CS-NPs. For **b** and **c**: Blue and Red solid lines belong to seeds and core-shell nanoparticles, respectively. Blue and Red dashed lines belong to standard Pd and Pt, respectively. Scale bar in **a** is 10 nm^{-1} .

Second, we performed atomic-resolution HAADF-STEM imaging on our CS-NPs and calculated the strains using geometric phase analysis (GPA). Fig. R5a&d show clear core and shell boundary in 2D which allows us to do GPA on two single-crystalline TO (truncated octahedron) CS-NPs. The GPA results show the strains are less than 5%

for most of the particles. It's notable that the apparent strains on the edge are artifacts in the GPA analysis, rather than being an intrinsic material feature (*Ultramicroscopy* **157**, 91-97(2015)).

Fig. R5 | Geometric phase analysis of TO Pd@Pt nanoparticles. **a** and **d** are atomic-resolution HAADF-STEM images of two truncated octahedron shaped Pd@Pt nanoparticles showing [100] zone axis. **b**, **c**, **e** and **f** are corresponding strain field (ϵ_{xx} and ϵ_{yy}) mapping from -10 % to 10 %. Scale bar in **a** and **d**, 2 nm.

Third, we compare our measurement with the results reported in literatures. Indeed, some slices of our 3D strain mapping (Supplementary Figs. 11-13) are similar to previous work (*Nat. Commun.* **13**, 5957 (2022)) which exhibit the strain distribution correlated to core-shell structure (Fig. R6). Both of them show a relatively low field strength, compared to works reported using GPA analysis. That is because the accuracy of GPA analysis is disturbed by the overall background strength and noise, which introducing artifacts into lattice strain measurements (*ACS Catal.* **10**, 5529–5541(2020)). Although the measured strain is small (Fig. R6), we have emphasized this result in our updated manuscript (page 9).

Fig. R6 | Comparison of the strain in this work and recent published works. a shows strain slices corresponding to core-shell interfaces. **b** is adopted from previous work in literature (Y. Yang et al. *Nat. Commun.* **13**, 5957 (2022)). Color range in **a** and **b** are both from -2.5% to 2.5%.

We added how to measured atomic displacements, strains and mean bond lengths in the Methods section.

Comment 6: “{111} and {200} surfaces have different surface energies. This suggests that the thickness of intermixed layer can be different depending on the facets. Any comment on this?”

This is a good point. We analyzed the diffusive interfaces on different crystal faces. Fig. R7 show the radially-averaged Pd concentrations along the core to the shell at {200} and {111} facets, respectively. No significant difference is observed for EPB and TO particles. As for PB particle, Pd concentration along {200} facets drop faster than along {111} facets. This is because the pentagonal bipyramid particle contains ten large {111} facets with five narrow {200} facets at the edge of two adjacent {111} facets on the same crystalline domain. The number of atoms along {200} facets in the interface were much smaller than that of {111}, leading to a larger descent slope for the Pd concentration curves with a larger error.

Fig. R7 | Radially averaged Pd concentrations along {111} and {200} directions for three particles.

Comment 7: “The authors presented three atomic maps of core-shell nanoparticles, but I think the 3D maps are not validated enough. Here, I suggest some methods to validate the maps:

- 1) Simulate STEM images from reconstructed maps, and compare them with the original ones;*
- 2) Calculate the 3D precision (root-mean-square-deviation) by re-reconstruction. (Please refer to [Y. Yuan et al., Nature Materials 21, 95-102 (2022)] or [B. H. Kim et al., Science 368, 60-67 (2020)])”*

Thanks for pointing this out. We performed HAADF-STEM simulations using the multi-slice method and estimated the precision of atomic coordinates by AET reconstruction as the literatures did (*Nature* **570**, 500-503 (2019); *Nat. Mater.* **21**, 95-102 (2022)). The validation procedure is listed below:

- 1) The multi-slice simulations of tomographic tilt series from PB, EPB and TO models were performed under the same conditions as the AET experiments: 300 keV acceleration energy, -398 nm C3 aberration, 414 μm C5 aberration, 30 mrad convergence semi-angle, and 39.4 mrad and 200 mrad HAADF detector inner and outer semi-angles. 4 frozen phonon configurations and 2 \AA slice thickness was applied for the simulations.
- 2) Using the same reconstruction RESIRE algorithm, 3D reconstructions were obtained from simulated tilt series images. The atomic coordinates of three particles

were traced from corresponding 3D reconstruction volumes using the same procedure described in the Methods section. We identified common atom pairs between the atomic coordinates traced from multi-slice simulated tomograms and the experiment models using half of the identical fcc Pd-Pd bond length (1.38 Å) as a threshold distance. RMSD of all common atom pairs were 19 pm (PB) with 99.8% atoms traced, 14 pm (EPB) with 99.9% atoms traced and 17 pm (TO) with 99.8% atoms traced, respectively (Fig. R8).

The 3D precision (~20 pm) and tracing error are similar to the previous reported results (*Nat. Mater.* **21**, 95-102 (2022); *Nature* **542**, 75-79 (2017)). We have added this validation in the main text (page 5) and Methods section of the revised manuscript; and a new supplementary Fig. 7 has been updated.

Fig. R8 | Multi-slice simulation of three core-shell nanoparticles. a-c, Multi-slice simulation images calculated from the experimental 3D atomic model of for PB, EPB and TO particles, respectively. They are in good agreement with denoised experimental images. d, Histogram of the root-mean-square deviation (RMSD) between the experimental 3D atomic models and the new 3D atomic models traced from simulated reconstructions. Scale bar in a-c, 2 nm.

Comment 8: “In ‘Methods’ section, the procedure for 3D atomic coordinates determination and chemical species classification is described. In the manuscript, there are some manual check and modifications as follows: “These positions were further checked manually to correct for unidentified or misidentified atoms due to fitting failure or a large chunk of connected intensity blobs from multiple atoms”; “we

performed local surface re-classification and manually checked a small percentage of the inconsistent atoms²⁷.” Could you describe more on those ‘manual check’? What are the criteria to determine ‘unidentified or misidentified atoms’ and ‘the inconsistent atoms’?”

We want to clarify this very important point. In the auto-tracing process, firstly we found all local maximums; and then excluded non-atoms based on two criteria:

- 1) The integrated intensity of a box centered on the local maximum position must exceed the threshold of noisy intensities.
- 2) The minimum distance between two reasonable atom sites must be greater than the reasonable lower limit of bond length.

After deleting all non-atom positions, we executed 3D polynomial fitting on atom positions to get sub-pixel resolution. About 99% of atoms were successfully traced within this procedure.

Because the 3D reconstruction usually suffers from the so-called “missing wedge” problem, the artifacts and noises in the final 3D tomography reconstruction would generate redundant local maximums on one atom area, as is shown in Fig. R9. Therefore, auto-tracing procedure will cause fitting failures and cannot yield a reasonable atom position when a large chunk of connected intensity blobs from multiple atoms present. In this case, both of them are not on the center position of this atom. One atom needs to be deleted first as its integrated intensity is not high enough, or all of them may be removed based on the minimum distance criteria. In either case, we cannot get a reasonable traced position of this atom automatically. These atoms are classified as “unidentified or misidentified atoms”. Therefore, we performed manual check after auto-tracing process as previous study suggested (*Nature* **542**, 75-79 (2017); *Nature* **570**, 500-503 (2019)).

Here is how we performed the manual check procedure (Fig. R9). We drew the isosurface of the whole 3D model, and checked if all atom areas are correctly traced one by one. For unidentified or misidentified atoms (either atoms were unphysically close together or a blob is missing an associated atom), we removed the wrong traced positions and traced them again using polynomial fitting. We added this into Methods section. It’s notable that manual check happens mostly in the missing wedge direction. For PB, EPB and TO particles, we manually added less than 1% atoms.

Fig. R9 | Principle of manual tracing. Green atoms are traced at right position. Red atoms are unidentified ones.

After tracing, we classified all atoms into Pd and Pt using K-mean clustering method. In HAADF-STEM imaging mode, the image contrast is Z contrast, approximately proportional to square of atomic number (Z). We were able to distinguish Pd ($Z=46$) and Pt ($Z=78$) based on the integrated intensity of voxels surround the center of an atom. Due to the missing wedge problem, the intensity of the surface atoms from missing wedge direction are weaker than the atoms insides. These atoms are misclassified to Pd when using the K-mean algorithm (*Nature* **592**, 60-64 (2021)). We found they are inconsistent atoms. Since our EDX mapping results clearly show Pt signal are dominate and there are no Pd atoms on the particle surface after the growth of a thick Pt shell (Fig. R11). We manually checked a small percentage of the inconsistent atoms on the surface and performed local surface re-classification (*Nature* **570**, 500-503 (2019)) at a lower intensity threshold to classify the surface Pt.

The details of manual tracing and classification above were updated in the Methods section of revised manuscript.

Comment 9: "In Fig. 2, the zone axes of (a) and (d) are different. I recommend to change Fig. 2d with an image with the same projection direction of Fig. 2a. Presenting the atomic distribution around the five-fold co-axis could be helpful for material scientist who are interested in polyhedral core-shell nanoparticles."

Thanks for pointing this out. We have replaced panel Fig. 2d as referee 1 suggested (Fig. R10) in revised manuscript.

Fig. R10 | New version of Fig. 2 with panel d replaced.

Response to referee 2:

Comment 1: *“For the core–shell nanoparticle growth, the shell growth is strongly reliant on core particle size and shape. Authors should provide the HRTEM/STEM of the as-synthesized core nanoparticle accompanied by the core–shell nanoparticle in Figure S1. Authors should indicate the size of core–shell nanoparticle in the 1st paragraph of the Results section.”*

Thanks for the good suggestion. We have performed more STEM imaging experiment and added the atomic resolution HR-STEM images of PB, EPB, TO CS-NPs and corresponding Pd seeds in updated supplementary Fig. 1 (also see Fig. R3). We added one sentence to indicate the size of TO CS-NPs (page 5).

Comment 2: *“On page 5, the authors indicate that “Despite of the variety in size and shape, the interfaces between the Pd core and the Pt shell show diffuse chemical profiles, without a defined sharp boundary in chemical composition for all three particles.” In fact, the core–shell nanoparticle synthesized in this manuscript is around 8.7 nm, 6.6 nm, 7.5 nm, the word “variety” can not be used here. In a representative example [J. Am. Chem. Soc. 2015, 137, 47, 15036–15042], Pd@Pt nanoparticle is synthesized with 20 nm size, but nanoparticle > 10 nm is not included in this study. Therefore, authors may not draw the conclusion in the Introduction part “core-shell interface is atomically diffuse with an average thickness of 4.2 Å, irrespective of the particle’s size, morphology or crystallographic texture”.”*

Thanks for pointing it out. We have removed the size dependence in our revised manuscript.

Comment 3: *“In Figure 2 (d), EPB (e) and TO (f), the authors use the red and blue dots to show the Pd and Pt atoms, respectively. We notice that the Pd core is enclosed in a Pt shell, with some Pd atoms diffuse across core–shell interface, but there is no Pd atom at the surface. Because the catalyst properties are highly correlated to the surface atom, and authors also predict that “atomically diffuse feature affect applications of core-shell structures in catalysis.” We suggest authors perform EDX elemental mapping for a single Pd@Pt nanoparticle to confirm that the surface nanoparticle is pure Pt without Pd.”*

We did EDX mapping on single TO Pd@Pt nanoparticles, most of which were oriented along the [100] zone axis, which facilitates atomic resolution EDX analysis. Fig. R11a & c show the atomic resolution HR-STEM images and corresponding EDS maps of two individual TO Pd@Pt nanoparticles, respectively. We did line-cut of (a) and (c) along the yellow arrows and the line profiles are drawn in Fig. R11b&d, showing a strong Pt signal in the shell especially at the edge layer/layers of shell (blue

arrows). We confirmed that the surface of CS-NPs is pure Pt without Pd.

Fig. R11 | EDX mappings and line profiles of single Pd@Pt core-shell nanoparticles. **a** and **c** are EDX mapping of two TO shaped Pd@Pt nanoparticles. Yellow arrow in HAADF images show the sampling position of line profiles below. **b** and **d** are line profiles of two nanoparticles. Blue arrows indicate the surface of Pd@Pt nanoparticles is pure Pt. Scale bar in **a** and **c**, 2 nm.

Comment 4: “On page 4, the authors indicate that “The radially-averaged concentration of Pd in the interface drops in a two-step manner, which is correlated to the electric double layers.” The author should provide definitions here to explain the terminology “electric double layers”.”

We added one sentence to explain the terminology “electric double layers” as referee 2 suggested in the Methods of our revised manuscript. One reference is also cited to explain it.

Comment 5: “In addition, on page 14 and Figure 4, the authors indicate that “A few surface Pd atoms was [should be “were”] ionized and formed an electric double layer around the seed.” This is not supported by solid data in this manuscript. In the conclusion section, the authors indicate that “Our results suggest that the diffuse interface is highly related to the electric double layers of the seeds, which could possibly be tuned by manipulating the Debye length.” However, the authors did not show related data and even the method for tuning the Debye length.”

Thanks for pointing it out. To support our hypothesis, we performed more atomic resolution STEM imaging using cryogenic temperature holder. Please refer to reply 2&3 to referee 1 (Fig. R1-2). We have observed individual free Pd atoms in Pd seeds; we observed the intermediate products during the shell growth; single atoms and small clusters are observed. We have added these results in our revised manuscript (page 15). These free Pd atoms and small clusters could possibly form electric double layers around the Pd seeds, which could be tuned by changing ionic strength (salt concentration) or relative permittivity (solvent). We clarify this in the revised manuscript.

Response to referee 3:

Comment 1: “The authors conclude from Fig. 3d and Supplementary Figs. 12a-b that, isolated Pd atoms in the Pt shell are far more common than isolated Pt atoms in the Pd core. If true, this would be somewhat consistent with the proposed growth mechanism. However, please note that Fig. 3d and Supplementary Figs. 12a-b are distributions of isolated Pd atoms only outside of a 3D pure Pd mask. The comparison to isolated Pt distribution is not clear since these data are not given.”

This is a very good point. We followed the referee 3’s suggestion and investigated the distributions of isolated Pt in the Pd core. As one can see from Fig. R12, comparing to Fig. 3d and Supplementary Figs. 16a-b, isolated Pt atoms are less common in the Pd core. In particular, there is no isolated Pt atoms in TO particles. We added these results in our revised manuscript (updated Supplementary Fig. 15).

Fig. R12 | Rendering of diffused Pt atoms in Pd core. **a**, total view of diffused Pt atoms in Pd core for PB, EPB and TO particles, respectively. Surface of the pure Pt shells are colored cyan. **b**, cut view of **a**.

Comment 2: “Also, within the confinement of this discussion the shell and core need to be better defined. The 3D mask is generated under a subjective constrain (according to the Method) that CN of Pd-Pd was not less than 12, therefore, it’s a clean Pd core without any adjacent Pt atoms by subjective choice of atomic segregation. A similar 3D mask could be made for Pt and end up with a clean Pt shell as well. Therefore there should be 3 layers using this approach: pure core, diffusion layer and pure Pt shell. If they begin the data analysis with Pt, an opposite conclusion could be drawn. The suggestion that Pt is far less likely to diffuse into Pd core is not conclusive based on these arguments yet, at least not from a logical stand point of view.”

This is a very important point we want to clarify. As shown in Fig. R13, the asymmetric diffuse interface is observed no matter whether we choose to start at a pure Pd core or a pure Pt shell: the Pt concentration drops over a short distance into the Pd core, while the Pd concentration drops more slowly into the Pt shell. We want to clarify that in both analyses, the Pd concentrations drops quickly to 5%, and then at a slower rate until reaching 0%. The concentration distribution of Pd leading into the Pt shell has a longer tail than the concentration distribution of Pt leading into the Pd core, which means there are more isolated Pd atoms in Pt shell than isolated Pt atoms in Pd core (Fig. R12, Fig. 3d and Supplementary Fig. 15).

Fig. R13 | Radially-averaged concentration of element Pd for three particles. a, Concentration line got when exfoliation started from pure Pd core. **b,** Concentration line got when exfoliation started from pure Pt shell.

Comment 3: “Page 12, second paragraph: “The same atomic slice in both models shows that the diffused Pd atoms in the Pt shell remain the same position and chemical environment (Supplementary Fig. 13, the yellow circles).”

The authors concluded that the diffused Pd at the interface does not originate from the atomic motion due to heating during the synthesis or baking process. However, it's unclear how thoroughly the influence of heating is examined. The discussion and figure provided seem to indicate that only one slice of each of the reconstructed 3D model from one particle before and after 48 hours of 180 °C baking is examined. If this is the case, it would be a very small sampling size to claim that the diffuse behavior remains the same for one entire particle, let alone for all particles. Also, it is not clear in the manuscript if the particle had undergone baking at 180 °C for 24 hrs in high vacuum (described in methods) before the 48 hour baking. Have the authors considered the possibility that the diffusion being initiated by baking, but stabilized after a given amount of time under constant temperature, or only when baking in solution?”

We want to clarify this important point. We have compared 3D atomic coordinates and species of all 6000 atoms in the same pentagonal bipyramid shaped particle before and after baking. By comparing their 3D atomic coordinates, we confirmed that 85.7% of total atoms were consistent between the two models (the threshold is half of the mean bond length, 1.38 Å) and the precision of our 3D atomic structure

determination method is 70 pm. The chemical accuracy of consistent atom pairs in two reconstruction was 97.5%.

The position precision is less accurate than the one without any baking reported in literature (25 pm in 2019 Nature (*Nature* **570**, 500-503 (2019))), indicating there is some lattice relaxation during the baking. Fig. R14 shows the 3D deviation maps between the coordinates before and after baking, indicating a lattice relaxation during the baking, which happened mostly along the fivefold axis direction (Z direction in Fig. R14b) where the distortion was highest.

Although the surface atoms which could move their positions more easily by surface reconstruction (*J. Am. Chem. Soc.* **139**, 4551–4558 (2017)), we found 90.9% atoms in the bulk of the particles ($CN \geq 11$) were consistent between the two models, with a chemical consistency of 97.8%, as high as AET can possibly yield (*Nature* **570**, 500-503 (2019); *MRS Bull.* **45**, 290-297 (2020)). Overall statistics of all atoms show the baking at 180 °C cannot trigger a dramatic evolution due to atomic diffusion. We also show more slices comparing the atom species and their chemical environment (Fig. R15).

Fig. R14 | Comparison of atom coordinates before and after baking for a pentagonal bipyramid shaped Pd@Pt core-shell nanoparticle. a, Segmentation of the fivefold symmetric CS-NP into five regions. **b,** 3D displacement map of atoms before and after baking for five regions.

Fig. R15 | Atomic slices of a pentagonal bipyramid shaped Pd@Pt CS-NP before and after baking. Yellow circles highlight consistent diffuse interface areas. Scale bar is 2 nm.

We investigated the influence of heating on Pd@Pt CS-NPs in the literature. *In-situ* heating experiments have shown that the CS-NPs remained almost identical below 400 °C; only a small portion of surface atoms become mobile enough to diffuse at 400 °C (*ACS Nano* **11**, 4571-4581 (2017)). It is also notable that simulations have been done in literatures to study the energy barrier to swap a Pd with Pt in CS-NP system (*J. Phys. Chem. C* **116**, 8664–8671(2012); *ACS Nano* **11**, 4571-4581 (2017)); the energy way is too high for a direct atomic diffusion in Pd@Pt at 180 °C. Based on these results, we think the thermal diffusion of Pd-Pt in solution is rare at such a low temperature. We have added these results in our revised manuscript.

Comment 4: “Page 13, “The size distributions between Pd seeds and the final Pd cores in CS-NPs (Supplementary Figs. 1d-f) are almost identical, indicating that the etching of Pd seeds with Pt(IV) was negligible. The diffuse Pd atoms in the interface cannot solely come from the galvanic replacement of the Pd seeds with Pt(IV) precursors under a reducing environment, as this would result in a smaller Pd core” Based on the data the thickness of the diffusion layer is ~4.2 Å. Base on the concentration distribution curve, the etching depth, if existed, seems to be slightly less than ~ 0.2 nm. The minimum scale of the particle size statistics in supplementary Figs. 1d-f is about 0.5 nm. The resolution of the particle size statistics is maybe enough to claim that any etching of Pd larger than 1nm is negligible, but not enough to show that a 0.2 nm deep surface etching did not happen. Therefore, this data is not enough to support the claim that the diffused Pd is not solely coming from the galvanic replacement of the Pd seeds.”

This is a very good point. We acquired more than fifty large-scale atomic resolution HAADF-STEM images of both seeds and corresponding CS-NPs. In these images, we were able to get a much finer size comparison between the seeds and the core in CS-NPs. We use TO seeds and core-shells for statistics, since there are all in single

crystalline fcc structure with rectangles shapes, which are easy to get zone axis images. Most of them were oriented along the [100] zone axis. We summarized the length and width of TO seeds and the core of CS-NPs separately from more than 200 particles by counting the number of atom columns in atomic resolution images (Fig. R16). If the particles were tilted such that only lattice fringes were visible, we measured size only along the direction with atomic resolution. The spacing of [200] planes for fcc Pd and Pt is 0.194 nm and 0.196 nm, respectively. Therefore, we have estimated the size distribution of seeds and core-shells with a precision better than 0.2 nm.

Fig. R16 | Procedure of counting the size of TO seeds and core-shells. **a**, An atomic resolution HAADF-STEM image of TO seeds. All nanoparticles were oriented to [100] zone axis. Red lines circled particles that were completely captured in the scan field of view. **b**, particles automatically recognized by code. **c**, a representative rectangle Pd seed with 33 rows and 36 columns of atoms.

Fig. R17 shows the particle size statistics, indicating any etching of Pd larger than 0.2 nm is negligible. It is been reported that the galvanic replacement of the Pd seeds would yield smaller Pd particles in the final CS-NP system (*Adv. Mater.* **25**, 6313-6333 (2013)), which we have not observed at a resolution of 0.2 nm. We have added these results in the revised manuscript (page 17-18).

Fig. R17 | Size statistics of TO seeds and core-shells. **a**, Representative atomic resolution HAADF-STEM images of cuboid Pd seeds in large scale for statistics. **b**, Representative atomic resolution HAADF-STEM images of Pd@Pt CS-NPs for

statistics. **c**, particle size statistics of TO seeds, core and whole of TO CS-NPs. Scale bar in **a** and **b**, 5 nm.

Comment 5: *“Unless these concerns are addressed, the only result supporting the proposed mechanism seems to be the subjective two step declining pattern of the Pd concentration curve. It will then need to be compared with all other possible growth mechanisms to eliminate their possibility of a similar or better match.”*

We want to clarify this important point. We have proved that three major possible sources of Pd atoms cannot provide enough Pd to form the diffusive layers we observed in 3D during shell growth. These include temperature-induced atomic diffusion, galvanic replacement of the Pd seeds, and the residual Pd salts from seed-growth. We have provided atomic resolution images showing two major results, (1) there are free Pd atoms in the well-washed seeds solution before adding Pt precursors (Fig. R1); (2) there are free Pt/Pd atoms during the shell growth (Fig. R2). This is a direct evidence validating our proposed mechanism in revised Fig. 5. To avoid any confusion, we clarified these points in our revised manuscript (page 14-15).

Comment 6: *“Abstract: Instead of forming an atomically-sharp boundary, the core-shell interface is atomically diffuse with an average thickness of 4.2 Å, irrespective of the particle’s size, morphology or crystallographic texture. This claim is misleading. Analysis of diffusive core-shell interfaces was done on three different particles. Size difference of these three particle does not cover the entire possible size distribution spectrum. Claiming from this result with three data points that the diffuse layer is constant irrespective of size seems premature.”*

Thanks for pointing this out. To avoid any confusion, we have removed the size dependence in our revised manuscript.

Comment 7: *“Method: The Pd(II) solutions were heated at a certain temperature to yield Pd seeds with different morphologies. Please specified the temperatures used.”*

We have clarified the temperatures we used in Pd seed synthesis in updated Methods.

Comment 8: *“Supplementary Fig. 1f How is the diameter defined for non-spherical particles?”*

We use the side length of rectangles as the diameter of TO particles (Fig. R16). Two different side lengths are treated as two values in the statistics, since TO seeds and core-shells in STEM images are all cuboids or truncated octahedrons sited on carbon films with random orientations.

Comment 9: *“Page 4: The radially-averaged concentration of Pd in the interface*

drops in a two-step manner, which is correlated to the electric double layers. While the diffusion behavior and the potential near the electric double layer could be somewhat correlated, it's a stretch to conclude from observation of the radially-averaged concentration curve that the concentration drops in "two" steps."

Thanks for this good point. We enlarged the averaged concentration curve of Pd in Fig. R18, which shows asymmetric: The Pd concentration distribution around 5% (near Pt shell) has a longer tail than the Pd concentration distribution around 95% (near Pd core). We believe it would be more precise to say that there is a “two-sided distribution”, with a smaller interface width for Pt concentration in the Pd core and a larger interface width for the Pt concentration in the Pt shell. We removed “two-steps manner” and modified the description of this asymmetric Pd concentration curve in the main text (page 4 and page 14; Supplementary Fig. 15).

Fig. R18 | Mean Pd concentrations along the core to the shell. Radially-averaged Pd concentrations along the core to the shell for three particles, where the diffuse interface is segmented to three regions based on averaged Pd concentration: from 99% to 95% (highlighted with yellow band, 0.6 Å), from 95% to 5% (highlighted with green band, 4.2 Å), from 5% to 1% (highlighted with blue, 4.1 Å). We chose the green band as the range of diffuse interfaces where Pd concentration drops fast from the core to the shell. Zero depth was defined as where Pd concentration equals to 50%. The yellow and blue band represent two areas where Pd concentration decreases slowly. Most of isolated Pd atoms distribute in the blue area. The blue band is far wider than the yellow band, showing the asymmetric feature of interface mixing.

Comment 10: “*Methods: Analysis of diffusive core-shell interfaces.*

Could you explain why a fixed distance is used to exfoliate the 3D mask for the core instead of adjacent atomic layers? For different surfaces, the distance between

adjacent atomic layers would be different. By fixing the distance to exfoliate, it looks like the exfoliated layer may be centered for the atoms of 1 surface layer but will cut into the edge of atoms on a different surface or may never intercept with the atomic center. How is number of atoms determined for each layer if this is the case?"

Fig. R19 | Pd concentrations of three particles calculated with different exfoliation thickness. Voxel size is 0.343 Å for three particles.

This is an important point we want to clarify. Since the crystal plane spacing is different for [200], [111] and other high-index crystal planes, it is difficult to describe the average thickness of diffused Pd-Pt interface if we exfoliate the nanoparticle directly by atomic layers. Instead, we used a fixed distance instead of atomic layers so

that the thickness of each exfoliated surface layer is identical. We attributed atoms located at the edge of two layers to the nearest layer instead of cutting them to both layers. As shown in Fig. R19, we changed the thickness of each layer in the exfoliation using different voxels; it doesn't have much influence on the Pd concentration curve. Obviously, while larger exfoliation thickness resulted in a smoother curve, smaller thickness means a finer characterization of important features on the Pd-Pt interface. Since the voxel size is 0.343 Å, the mean spacing of [111] and [200] planes for three particles were 2.25 Å (6.6 voxels) and 1.95 Å (5.7 voxels), respectively. In order to sample the Pd concentration with a fine step to the best we could, we finally chose the exfoliation thickness about two times smaller than the mean spacing of crystal planes, i.e., 3 voxels (1.03 Å).

Comment 11: “*Method: Platinum salt solutions were added into pre-heated (180 °C) oleylamine solutions containing different seeds to grow the Pt shell. The solutions were aged at 180 °C for 1 hrs. 1 hrs should be 1 hr.*”

Thanks for pointed it out. We have corrected it in our revised manuscript.

Reviewers' Comments:

Reviewer #1:

Remarks to the Author:

Referee 1:

I appreciate the authors' times and efforts in addressing my comments, many of which are resolved properly. I support publishing this manuscript in Nature Communications but I would like to point out some minor aspects which can be included in the revised manuscript.

1. One remaining issue in the revised manuscript is related to the Debye length: for me, it is strange to define the Debye screening length in the organic medium (e.g. oleylamine). To my knowledge, Debye length is defined in electrolytes where ions are stabilized to exist; there would be no Debye length or EDLs in organic solvent. The Pd species dissolved from the seed are thus supposed to be present as free atoms (with no charge) and/or ions coordinated with ligands (making ion-ligand complexes electrically neutral).

2. In the upper panel of Fig. S9b, the colors of the two plots seem to be reversed.

3. Could the authors include a description of the thicknesses of the intermixed layers along {111} and {200} that are similar to each other in the manuscript? This could be an intriguing topic to explore and would add value to the paper.

4. I recommend the authors to revise the whole manuscript once more to make it more naturally. For example, in the abstract, the sentence "We observed dissolved free Pd and Pt single atoms and sub-nanometer clusters using cryogenic electron microscopy", which may be added in response to my previous comment, is awkward considering the connection with the previous sentence.

Reviewer #2:

Remarks to the Author:

The authors have made appropriate revisions according to my comments. I think the paper can be accepted for publication.

Response to the referees:

We thank all three referees for reviewing our manuscript again and making very good suggestions, which have been very helpful to us to make improvements to the manuscript. Below is a point-by-point response to the referee 1' comments and suggestions. All the revisions made in the manuscript are marked blue.

Response to referee 1:

Comment 1: “One remaining issue in the revised manuscript is related to the Debye length: for me, it is strange to define the Debye screening length in the organic medium (e.g. oleylamine). To my knowledge, Debye length is defined in electrolytes where ions are stabilized to exist; there would be no Debye length or EDLs in organic solvent. The Pd species dissolved from the seed are thus supposed to be present as free atoms (with no charge) and/or ions coordinated with ligands (making ion-ligand complexes electrically neutral).”

We thank referee 1 for raising this question about whether we can use electric double layers (EDLs) model to describe the solubility of noble metal salts in organic solvents like oleylamine. We agree it is rare to calculate Debye length in nonpolar solvent. The relative permittivity of oleylamine is relatively small ($\epsilon_r \approx 3.1$) compared to most polar solvents (see Table 3.2 in *Intermolecular and Surface Forces 3rd Ch. 3* (Elsevier, 2011)), it belongs to weak-polar solvent. After carefully consider the length of oleylamine and the steric hindrance, one layer of oleylamine could occupy more than 10 Å if the orientation is right. We follow referee 1's suggestion to remove the discussion of EDL in the revised manuscript. We will study this effect in aqueous solution in a separated paper.

We also would like to point out that it's been reported in many studies that the EDLs could form in polar organic solvents where the electrolytes are soluble such as ionic liquids or amide (*J. Phys. Chem.*, **73**, 11, 3598–3608 (1969), *J. Electroanal. Chem.*, **588**, 285–295 (2006), *Phys. Chem. Chem. Phys.*, **12**, 5468-5479 (2010), *Chem. Rev.*, **122**, 10821–10859 (2022)). As in oleylamine solution, FTIR spectra of metal ions-oleylamine complexes clearly showed the existence of Cu(I) or Mn(II) ions in oleylamine stabilized by coordination (Fig. R1, *J. Phys. Chem. C*, **118**, 18, 9801–9808 (2014), *Nanoscale*, **6**, 5918-5925 (2014)).

[redacted]

Fig. R1 | Fourier transform infrared (FTIR) spectra of Cu(I) and Mn(II) solvated in oleylamine. **a**, FTIR spectra of oleylamine (black) and Cu⁺-oleylamine complexes (red). In the spectrum of Cu⁺-oleylamine complexes, the N-H stretching region shifts to 3230.5 cm⁻¹ and an obvious broadening in the bandwidth is observed. This panel is adopted from *J. Phys. Chem. C*, **118**, 18, 9801–9808 (2014). **b**, Photographs and FTIR spectra of pure oleylamine (OM) and solutions 1–5. The five solutions were formed by firstly mixing Mn(Ac)₂ and oleylamine at 80 °C, and then heating the mixture at 100 °C. All spectra were normalized according to the absorbance at 2922 cm⁻¹. The -NH₂ symmetric stretch vibration of oleylamine at 3318 cm⁻¹ is greatly reduced in intensity compared to pure oleylamine. This panel is adopted from *Nanoscale*, **6**, 5918-5925 (2014).

Nevertheless, we could not exclude the possibility that Pd species dissolved from the seed may be presented as free atoms, which we could not distinguish directly in our TEM experiments (as shown in Fig. 4). We thank referee 1 for raising up this hypothesis and include it in page 19 of the revised manuscript.

Comment 2: “*In the upper panel of Fig. S9b, the colors of the two plots seem to be reversed.*”

We double checked the plots and would like to clarify this point. We can notice that in the radial averaged intensity profiles of SAEDs, the unit of X axis is 1/nm. The larger the reciprocal space vector g (1/nm), the smaller the corresponding real space lattice spacing. Since the lattice constant of fcc Pd is smaller than fcc Pt, in reciprocal space the diffraction ring position of fcc Pd should be larger than that of fcc Pt, which is consistent with the line profiles in Supplementary Fig. 9b.

Comment 3: “*Could the authors include a description of the thicknesses of the intermixed layers along {111} and {200} that are similar to each other in the manuscript? This could be an intriguing topic to explore and would add value to the paper.*”

We thank referee 1 for this important suggestion. We have included this result and added three more panels as Supplementary Figs. 15g-i in our revised manuscript.

Comment 4: *“I recommend the authors to revise the whole manuscript once more to make it more naturally. For example, in the abstract, the sentence “We observed dissolved free Pd and Pt single atoms and sub-nanometer clusters using cryogenic electron microscopy”, which may be added in response to my previous comment, is awkward considering the connection with the previous sentence.”*

We have revised the writing of our manuscript to make it more naturally according to the suggestion.

Reviewers' Comments:

Reviewer #1:

Remarks to the Author:

All of my concerns are addressed in the revised manuscript. I now support a publication of the manuscript.